# Interannual increase of secondary organic aerosol in an urban environment

Marta Via[1,2], María Cruz Minguillón[1], Cristina Reche[1], Xavier Querol[1], Andrés Alastuey[1]

[1]Institute of Environmental Assessment and Water Research (IDAEA-CSIC), Barcelona, 08034, Spain

[2]Department of Applied Physics, University of Barcelona, Barcelona, 08028, Spain

*Correspondence to:* M. Via ([marta.via@idaea.csic](mailto:marta.via@idaea.csic)), M. C. Minguillón ([mariacruz.minguillon@idaea.csic.es](mailto:mariacruz.minguillon@idaea.csic.es))

**Abstract**. The evolution of fine aerosol ($PM_1$) species as well as the contribution of potential sources to the total organic aerosol (OA) at an urban background site in Barcelona, in the western Mediterranean basin (WMB), was investigated. For this purpose, a Quadrupole Aerosol Chemical Speciation Monitor (Q-ACSM) was deployed to acquire real-time measurements for two one-year periods: May 2014 - May 2015 (period A) and Sep 2017 - Oct 2018 (period B). Total $PM_1$ concentrations showed a slight decrease (from 10.1 to 9.6 µg·m$^{-3}$ from A to B), although the relative contribution of inorganic and organic compounds varied significantly.

Regarding inorganic compounds, $SO_4^{2-}$, black carbon (BC) and $NH_4^+$ showed a significant decrease from period A to B (-21%, -18%, -9%, respectively), whilst $NO_3^-$ concentrations were higher in B (+8%). Source apportionment revealed OA contained a 46% and a 70% of secondary origin (SOA) in periods A and B, respectively. Two secondary oxygenated OA sources (OOA) were differentiated by their oxidation status (i.e. aging): less-oxidized (LO-OOA) and more-oxidized (MO-OOA). Disregarding winter periods, when LO-OOA production was not favoured, LO-OOA transformation into MO-OOA was found more effective in period B. The lowest LO-OOA-to-MO-OOA ratio, excluding winter, was in September-October 2018 (0.65), implying an accumulation of aged OA after the high temperature and solar radiation conditions in the summer season. In addition to temperature, SOA (sum of OOA factors) was enhanced by exposure to a $NO_x$-polluted ambient and other pollutants, especially to $O_3$ and during afternoon hours. The anthropogenic primary OA sources identified, cooking-like OA (COA), hydrocarbon-like OA (HOA), and biomass burning OA (BBOA), decreased from period A to B in both absolute concentrations and relative contribution (as a whole, 44% and 30%, respectively). However, their concentrations and proportion to OA grew rapidly during highly-polluted episodes.

The influence of certain atmospheric episodes on OA sources was also assessed. Both SOA factors were boosted with long and medium-range circulations, especially those coming from inland Europe and the Mediterranean (triggering mainly MO-OOA) and summer breeze-driven regional circulation (mainly LO-OOA). In contrast, POA was enhanced either during air-renewal episodes or stagnation anticyclonic events.

## 1. Introduction

Fine particles ($PM_1$, those with aerodynamic diameter <1 µm) have a significant impact on human health (Trippetta et al., 2016; WHO, 2016; Yang et al., 2019), climate (Shrivastava et al., 2017), and visibility (Shi et al., 2014). Organic Aerosol (OA) is the main constituent of fine aerosol in the atmosphere (Zhang et al., 2007) and it can be classified regarding its origin as primary OA (POA), consisting of directly emitted OA; or secondary OA (SOA), resulting from chemical transformation of

pre-existing particles, nucleation or gas-to-particle condensation. Contributions to OA are still not fully understood due to large variability of their fingerprints, response to atmospheric dynamics and transport and evolution processes dependent on site-specific meteorological characteristics and precursors provision. Field-deployable aerosol mass spectrometers have been widely used to assess these variations (e.g. Jimenez et al. (2009)) and POA sources identification; such as hydrocarbon-like OA (HOA), biomass burning OA (BBOA) and cooking-like OA (COA). The fraction of organic mass measured at *m/z 44* (f44), typically dominated by the $CO_2^+$ ion and related to oxygenation, and at *m/z 43* (f43), dominated by $C_2H_3O^+$, can contribute retrieving information about ageing and oxidation state of SOA (Canagaratna et al., 2015). In Zurich, for instance, f44 during summer afternoons, when photochemical processes are most vigorous as indicated by high oxidant – OX ($O_3$ + $NO_2$), was found similar or lower than f44 on days with low – OX, while f43 (less oxidized fragment) tended to increase (Canonaco et al., 2015). The SOA, also referred as Oxygenated OA (OOA), is often divided into two factors: less-oxidized oxygenated OA (LO-OOA) and more-oxidized oxygenated OA (MO-OOA).

The atmospheric dynamics of the Western Mediterranean Basin (WMB) have been described in Millán (2014); Millán et al. (1997). The emissions of the densely populated, harbor-close, traffic-concurred and industrialized areas coupled to the breeze-driven regimes, complex topography and stagnation meteorological episodes prompt complex phenomena of transport and transformation. Previous studies in urban environments in the WMB have shown contributions of several sources to ambient $PM_{10}$ for long time periods (>10 years), such as road traffic exhaust, mineral, secondary nitrate and sulfate, marine aerosol, fuel-combustion and road dust resuspension or construction (Pandolfi et al., 2016). $PM_1$ composition and sources have also been studied, although with shorter time coverage (Brines et al., 2019; Pérez et al., 2008). Regarding the contribution to OA by its sources, previous studies in background environments in the WMB demonstrated the importance of OA in $PM_1$ (more than a 50%) in Ripoll et al., (2015b), 1-year study at a mountain site. In summer, most OA (90 %) consisted of oxygenated organic aerosol (OOA), split in LO-OOA and MO-OOA with a LO-OOA-to-MO-OOA ratio of 0.4, contrastingly to winter (71% of not-distinguished OOA). The marked diurnal cycles of OA components regardless of the air mass origin indicated that they were not only associated with anthropogenic and long-range-transported secondary OA but also with recently-produced biogenic SOA. In Minguillón et al. (2015), at a regional background site (Montseny) during a 1 year study, OA was also the major component of submicron aerosol (53% of $PM_1$), with a higher contribution in summer (58%, an 85% of which was OOA) than in winter (45%, with a 60% proportion of OOAs). The LO-OOA-to-MO-OOA ratio in summer was 0.9.

In urban environments in the WMB, information of OA sources was only available for relatively short periods of time. Previous high time resolution non-refractory $PM_1$ (NR-$PM_1$) chemical characterization and OA source apportionment consisted of two one-month campaigns in February-March 2009 (DAURE campaign Minguillón et al., 2011; Mohr et al., 2012, 2015) and August-September 2013 campaign (Minguillón et al., 2016). The LO-OOA-to-MO-OOA ratio was 0.90 in February-March 2009 *(*Mohr et al., 2012*)* and 1.14 in August-September 2013 (Minguillón et al., 2016), revealing different SOA oxidation states. Moreover, in Minguillón et al. (2016) the combination of [14]C analysis and OA source apportionment demonstrated that the enhanced formation of non-fossil secondary OA during the high traffic periods could be attributed to the reaction of BVOC (Biogenic Volatile Organic Compounds) precursors with $NO_x$ emitted from road traffic.

On the other hand, some studies have pointed out an increase of the oxidative potential of the atmosphere attributed to $NO_x$ reduction (Saiz-Lopez et al., 2017), which could have an impact on the oxidation state of SOA. Relatedly, Zhao et al. (2017) revealed the non-linear increasing relation of SOA production as a function of the NMOG-to-NOx precursors ratio (being NMOG the Non-Methane Organic Gases emitted by vehicles, which include VOC), entailing a coupled reduction of both pollutants has to be addressed in order to reduce SOA concentrations. Besides, Dai et al. (2019) highlighted the differences in conditions of Liquid Water Content (LWC), meteorological variables, and – OX concentrations for the SOA to take the aqueous-phase chemistry or photochemistry pathways depending on the season. For all that, there is a need of further

investigation about the oxidation state of OA in consideration of different seasons, years, meteorological scenarios and precursor levels to characterize these findings in an urban background site in the WMB.

This study aims to provide an intra and inter-annual analysis of OA sources along two one-year periods and an evaluation of differences and potential trends along these four years. To this end, a Quadrupole Aerosol Chemical Speciation Monitor (Q-ACSM) was deployed, together with complementary instrumentation, to comprehend site-specific nature and cyclicality of OA contributors. Accurate source characterization knowledge is necessary to design mitigating strategies of the effects of specific pollutants.

## 2. Methodology

### 2.1 Sampling site and period

The urban background site Palau Reial (PR; 41° 23′ 15″ N; 02° 07′ 05″ E; 80 m a.s.l.) is located in a residential area at the NW of Barcelona at 200 m distance from one of the most concurred avenues of the city (>60.000 vehicles per working day in 2014-

90 2018) (City council of Barcelona, 2019) (Fig. S1). Some recent studies in this site focusing on $PM_1$ and including organic speciation, present amongst others, hydrocarbon-like, cooking-like, biomass burning OA and diverse secondary aerosol sources (Brines et al., 2019; Minguillón et al., 2016). The atmospheric dynamics of the area are dominated by breeze regimes, consisting in a nocturnal NW wind component, a diurnal breeze development turning from SE to SW direction, and highest wind speeds around noon (Pérez et al., 2004).

Two intensive monitoring campaigns were carried out during May 2014 to May 2015 and September 2017 to October 2018 periods, which will be called hereinafter as period A and period B, respectively. Data availability for each measurement type is shown in Fig. S2. Henceforth, averages and data of any variable will correspond to the periods when Q-ACSM data is available.

### 2.2 ACSM settings, calibrations and data processing

A Q-ACSM (Aerodyne Research Inc.) was deployed at PR to measure NR-$PM_1$, distinguishing OA, $SO_4^{2-}$, $NO_3^-$, $NH_4^+$ and $Cl^-$ with 30-minute resolution. The Q-ACSM was connected to a general inlet with a 2.5 µm cut off and a flow rate of 3 L·min$^{-1}$, conducted through a nafion dryer maintaining the incoming RH below 40%. The instrument samples ambient air at 0.1 L·min$^{-1}$ through a critical orifice (100 µm in diameter) towards an aerodynamic lens which transmits particles between 75 and 650 nm (Liu et al., 2007). Particles are then flash-vaporized at 600°C in high vacuum conditions and ionized by hard-electron

impact (70 eV), and resulting fragments are analysed by a quadrupole mass spectrometer (Ng et al., 2011). The instrument is equipped with a filter-valve system, hence concentrations reported are the result of subtraction of particle-free to particle-laden signal. A fragmentation table (Allan et al., 2004), the ion transmission correction and the Response Factor (RF), are used to convert the signal spectra into organic or inorganic species concentrations. Ionization Efficiency (IE) and Relative Ion Efficiency (RIE) calibrations were conducted using 300-nm monodispersed $NH_4NO_3$ and $(NH_4)_2SO_4$ particles (Ng et al., 2011).

Final values for IE and RIEs for $NH_4^+$ and $SO_4^{2-}$, respectively, were 2.38·10$^{-11}$, 5.27, and 0.71 in period A; and 5.10·10$^{-11}$, 5.16, and 0.77 in period B. The significant difference between IE values is a consequence of the change of the detector before starting campaign B.

The Q-ACSM was operated with 24 scans per measurement (alternating sample/filter scan) with a scan speed of 200 ms·amu$^{-1}$, resulting in a 30-minute time resolution. Data acquisition software (versions 1.4.4.5, 1.5.2.1 and 1.6.0.0 depending on the

115 period) and analysis software (version 1.6.1.1) implemented in Igor Pro (WaveMetrics, Inc.) were provided by Aerodyne Research Inc. Data were corrected to account for flow rate changes and for response decay by using the $N_2$ signal. The

composition-dependent collection efficiency correction (CE) (Middlebrook et al., 2012) was applied for Period A and B respectively (minimum, maximum): (0.45,0.68), (0.50, 0.99), exceeding CE=0.6 a 0.13%  and a 1.5% of data).

## 2.3 Additional measurements and instrumentation

Black Carbon (BC) concentrations were measured with 1-minute time resolution by a Multi-Angle Absorption Photometer (MAAP, Thermofisher Model 5012). $PM_1$ concentrations were measured by summing up the particle size number distribution (ranging 12-470 nm) measured by a Scanning Mobility Particle Sizer (SMPS, TSI 3080) with 5-minute time resolution, and converted to mass concentrations by means of the composition-dependent aerosol density (DeCarlo et al., 2004). $SO_4^{2-}$, $NO_3^-$, $NH_4^+$ concentrations were determined in off-line $PM_1$ filters samples by ion chromatography and selective electrode methods. Organic Carbon (OC) concentrations were determined from off-line $PM_1$ samples by thermal-optical methods following the EUSAAR 2 protocol (Cavalli et al., 2010). Procedures of filter samples analysis are detailed in Supplementary Section 2.

Real time gaseous pollutants measurements, NO, $NO_2$ (Thermo Scientific, Model 43i), CO (Ecotech EC Model 9830), $O_3$ (SIR Model S5014) and $SO_2$ (Thermo Scientific Model 43C), operated by the Department of Environment of the Autonomous Government of Catalonia, were available at PR at a 30-minute resolution. Meteorological variables were provided by the Department of Meteorology of the University of Barcelona (Fig. S1) with a 10-minute time resolution. In all data analysis, time will always be UTC based.

Standardized protocols of quality control (COLOSSAL, COST Action CA16109, 2019) (in prep.) were carried out in both periods. The sum of all NR-$PM_1$ species (OA + $SO_4^{2-}$ + $NO_3^-$ + $NH_4^+$ + $Cl^-$) and BC was intercompared with co-located $PM_1$ measurements, assuming that the $PM_1$ contribution of the mineral and sea salt tail from the coarse PM or the trace elements are negligible. Also, species concentrations obtained by Q-ACSM were compared with the same components from off-line $PM_1$ determination except for Q-ACSM OA, which was compared with organic carbon (OC) (Section 3.1).

## 2.4 Source apportionment of OA

Source apportionment of the organic mass fraction was conducted applying the Positive Matrix Factorization (PMF) method (Paatero and Tapper, 1994) using the multilinear engine (ME-2) (Paatero, 1999). The SoFi (Source Finder) toolkit (Canonaco et al., 2013) version 6.8k developed by the Paul Scherrer Institute and Datalystica Ltd. was used. The PMF consists on the decomposition of the OA mass spectral matrix X, with m variables (m/z ions, columns) and n timepoints (Q-ACSM timestamps, rows). OA mass spectra was decomposed into two matrices, G and F, for a pre-set number of factors p by iteratively minimizing Q:

$$X = G \cdot F + E = \sum_{i,j}^{n,m} g_{ik} \cdot f_{kj} + e_{ij} \tag{1}$$

$$Q = \sum_{i,j}^{n,m} \left(\frac{e_{ij}}{\sigma_{ij}}\right)^2 \tag{2}$$

G is the contributions matrix with n time steps for p factors and F is the profile matrix of p factors with m m/z ions. The residual matrix E contains the unexplained fraction of X. In order to avoid local minima of the Q function, rotational tools are used to cover the whole m·n space. ME-2 provides control over the rotational ambiguity (Paatero and Hopke, 2009). A priori information can be introduced for some of the factors using the so-called a-value approach (Paatero and Hopke, 2009; Brown et al., 2012). The a-values range from zero to 1 and determine how much deviation from the anchor profile the model allows, with value zero meaning fully constrained.

The mass spectra used ranged from 12 to 120 Th and excluded higher m/z ions which accounted for a minor fraction of total signal (<3% on average), presented low S/N ratio or were interfered by the naphthalene signal. Each dataset was separated in four subperiods for source apportionment purposes to better capture the variation in secondary OA composition (reflected on

the oxidation state) and the sources only present part of the natural year. The subperiods used were: April-May, June-August, September-October and November-March. The selection of these subperiods, different from standard 3-months seasons, was carried out based on: i) variation of meteorological conditions (Fig. S9), ii) variation of source-specific markers f60 and f73, which are tracers for BBOA contribution, and iii) variation of the relation between f43 and f60, using monthly scatter plots, which may indicate the presence or absence of BBOA. The three criteria were met for a solution including a potential BBOA

for the November-March period. The subperiods will be hereinafter referred to as seasons.

The general steps followed to reach the presented OA source apportionment solution were those described by Crippa et al. (2013) and COLOSSAL guidelines (in prep.). Unconstrained PMF was performed with 3 to 8 factors. These runs were used to identify the present sources:  COA, HOA, BBOA, LO-OOA and MO-OOA. Subsequently, constraints were applied to the primary sources. PMF was run with 3 to 5 or 3 to 6 factors depending on the season. Differences between solutions of different

number of factors for each season are shown in Table S1 and Fig. S4. The solutions space was explored for HOA, COA and BBOA anchor profiles from Mohr et al. (2012), Crippa et al. (2013) and (Ng et al. (2010) whilst OOA factors were let be resolved freely. The criteria to choose the anchor profiles consisted of comparing correlations with external tracers or source markers: BC and $NO_x$ for HOA, $m/z55$ for COA, $m/z60$, $m/z73$ for BBOA, $SO_4^{2-}$ for MO-OOA and $NO_3^-$ for LO-OOA (Table S2). This led to the selection of the COA, HOA anchor profiles from Crippa et al. (2013) and the BBOA anchor from Ng et

al., 2010. All combinations of a-values within a range of 0 to 0.5, with steps of 0.1, were explored for each set of runs with the same number of factors, in order to select the most environmentally reasonable solutions. Optimization of the number of factors and a-value combinations implied considering: i) variation of the ratio between Q and Qexp (Qexp=m·n-p·(m+n)), which should present a steady descent from p to p+1, p+2 factors, being p the chosen number of factors; ii) correlation of time series contributions of OA factors with tracers; iii) scaled residuals of profiles and time series; iv) agreement between apportioned

and measured OA concentrations; v) gathered knowledge of site-specific atmosphere and potential sources.  In Table S1 the chosen solutions for each season are shown in contrast to solutions with one less and one more factor, all of them, with the optimal a-value combination according to correlation with external tracers. In almost all cases, it can be seen how the selected runs are the best compromise between the correlation with externals and the reliability of OA apportioned respect to the OA measured. Also, the Q/Qexp decrease was less steep from the selected run to the following compared to the previous to the

selected one. Nevertheless, in some cases, such as April-May 2018 or Nov-Mar 2014-2015, these figures were better for n-1 (or n+1) factors solutions, being n the number of factors chosen, but this extra (or lacking) factor allowed OOA differentiation (or differentiation incapacity), which provides a more accurate solution even in detriment of the aforementioned parameters. Moreover, in Figure S3, in most of the cases, the solutions for n factors showed the most reasonable compromise for scaled residuals along the time series and profiles in mainly all cases.

**2.5 Classification of atmospheric episodes or scenarios.**

Classification of atmospheric episodes was performed with the HYSPLIT model (Stein et al., 2015). Air mass back-trajectories for 120 h at three heights (750, 1500 and 2500 m a.s.l) were computed, with vertical flux modelling, for each day of measurements and interpreted to be classified regarding its predominant transport provenance into Atlantic North (AN), Winter Anticyclonic (WA) (from October to March), Europe (EU), Mediterranean (MED), North African (NAF) and Summer

Regional (SREG, from April to September), characteristics of which are discussed in previous works (Pey et al., 2010; Ripoll et al., 2014, 2015). In Figure S4 the relative contribution of each scenario per month is shown. The main difference between period A and B is on summer months; in period B, the main episode is SREG (always >60% of the days) whilst in A its proportion is reduced and more AN and AW episodes take place.

**3.  Results**

### 3.1 Comparison of Q-ACSM data with co-located measurements.

Comparison of Q-ACSM NR-PM$_1$ and MAAP BC with co-located PM$_1$ measurements can be found in Table S3 and Figure S5 a. The intercomparison correlation coefficients are $R^2$=0.61 and $R^2$=0.72 and slopes of the orthogonal distance fit 1.001 ± 0.006 and 1.177 ± 0.006 for periods A and B, respectively. The slopes are near one, especially in Period A. Nevertheless, SMPS underestimation of PM$_1$ could be partially attributed to differences in the particle size range measured, shifted to lower diameters for the SMPS (20 nm - 478.3 nm). Moreover, overestimation of primary OA sources by Q-ACSM by a factor of 1.2 to 1.5 has also been reported (Reyes-Villegas et al., 2018; Xu et al., 2018) as a consequence of source-unspecific OA RIE application, leading in turn to PM$_1$ overestimation by Q-ACSM.

The correlation coefficient for NR-PM$_1$ species off-line analysis of SO$_4^{2-}$, NO$_3^-$ and NH$_4^+$ is always above $R^2$>0.71, and also between OA and organic carbon (OC) ($R^2$=0.73 in A and $R^2$=0.86 in B) (Table S3, Figure S5 b). Slopes of the linear regression between Q-ACSM and off-line SO$_4^{2-}$, NO$_3^-$ and NH$_4^+$ are respectively 1.24, 1.90 and 1.73 for period A and 1.05, 1.76 and 1.68 for period B. The slopes of NO$_3^-$ are largely above 1, likely owing to volatilization artefacts in filters. The reason of the high slope for NH$_4^+$ is not clear, so neither determination problems in any of the filter species nor Q-ACSM overestimation cannot be completely discarded. The anion Cl$^-$ is not considered due to the very low concentrations and potential determination problems (Tobler et al., 2020).

OA-to-OC ratio is estimated from the slope in the scatterplot between these two variables, resulting in values of 2.69 and 2.94 in periods A and B, respectively, a 68% and 84% higher than the 1.6 value calculated with an AMS in Barcelona in the DAURE campaign (Minguillón et al., 2011; Mohr et al., 2015). The OM-to-OC ratio might have increased over the years since March 2009, which would be in accordance to an increasing SOA proportion found in these campaigns (2014-2015 and 2017-2018) respect to DAURE. Nevertheless, the values found in the present study are too high, as the most oxidized OOA expected values according both to chamber and ambient experiments should be around 2.3 (Canagaratna et al., 2015). This too high OA-to-OC ratios would point to filter artefacts caused by evaporation of semi-volatile OC compounds as well as similarly to the previous findings at Montsec (Ripoll et al., 2015) and Montseny (Minguillón et al., 2015), the aforementioned unspecific-source OA RIE overestimation for Q-ACSM. Even so, a steady increase of OOA oxidization over the years could be inferred due to the growth of this ratio from 2014-2015 to 2017-2018.

### 3.2 Submicron aerosol composition

Time series of co-located gases (Fig. S6) show rises of SO$_2$ in summer months likely related to shipping activity increase, O$_3$ increases during summer linked to higher photochemical enhancement and an increase of CO during cold periods, associated to shallower boundary layers. NO and NO$_2$ show similar behaviors of reduction towards warm months and sudden increases in cold periods.

There are evident differences in meteorological variables from one period to the other (Fig. S7). In summer A, temperature (24.4º C) and solar radiation (259 W) are lower and average relative humidity (71%) and wind speed (2.0 m/s) are higher than in period B (27.0º C, 280W, 70.0%,1.7%), indicating a probably rainier or cloudier summer in period A.

Data overview for periods A and B is shown in Table S4 and NR-PM$_1$ species and BC time series in Figure S8. Average PM$_1$ concentrations (± standard deviation) resulting from the sum of NR-PM$_1$ components and BC were 10.1 ± 6.7 µg·m⁻³ during campaign A and 9.6 ± 6.6 µg·m⁻³ during campaign B (variation of a 5% from A to B). A drop of a -5%, -21%, -9% and -18% is shown for of OA, SO$_4^{2-}$, NH$_4^+$ and BC, respectively, although NO$_3^-$ and Cl$^-$ had a positive variation of +8% and a +20% from period A to B. Previous data for PM$_1$, (2012-2018) showed a decreasing trend according to Mann-Kendall test (Figure S9) and a reduction over a longer period (2005-2017) was also determined for OC, EC. These long-term reductions could

imply that $PM_1$ and the OC and EC related pollutants (OA and BC) decreases found in this study reflect the tendency of the last years.

In both periods, OA was the largest contributor to $PM_1$, accounting for (average ± standard deviation) 43%±10% and 44%±17% of the total average mass (Q-ACSM NR-$PM_1$ species + BC), followed by $SO_4^{2-}$ (19%±11% and 18%±10%), BC (17%±9% and 13%±10%), $NO_3^-$ (10%±8% and 13%±10%), $NH_4^+$ (11%±8% and 11%±6%), and $Cl^-$ (0.4%±0.8% and 0.8%± 0.9%), respectively, for periods A and B.

### 3.2.1    Seasonal variation

Seasonally-averaged concentrations and time series of $PM_1$ and its components are displayed in Figure 1, and Figure S8. In period A, the highest concentrations of bulk $PM_1$ occurred in Sep-Oct (12.0 µg·m$^{-3}$) and the lowest were registered in summer (7.8 µg·m$^{-3}$). Contrastingly, in period B, submicron aerosol was maximum in summer (10.4 µg·m$^{-3}$) and minimum in winter (8.9 µg·m$^{-3}$). The differences in occurrence frequency of atmospheric episodes and data availability might be a direct cause of these mismatches of seasonal trends. The frequency of occurrence of episodes per month differs significantly from period A to period B (Fig. S4), concretely, in June, July and August the ratio of occurrence of SREG/AN is lower in period A than in period B (45% and 77% on average, respectively. This is coherent with lower $PM_1$ concentrations recorded in July A with respect to July B.

OA seasonal trend behaves similarly as bulk NR-$PM_1$ due to same causes, and photochemical enhancement in warm months due to higher sun irradiation. $SO_4^{2-}$ concentrations are in both cases higher during warmer months, opposite to the $NO_3^-$ concentrations trend (except for September-October 2014) as has been widely reported in previous studies (Minguillón et al., 2015; Pey et al., 2009; Ripoll et al., 2015). In period A, higher $NH_4^+$ concentrations happen during winter months whilst they are seasonally-stable through period B. BC shows the lowest concentrations in summer months in both periods (1.2, 1.3 µg·m$^{-3}$, respectively) more pronouncedly in period A.

### 3.3 OA source apportionment

COA accounts for a 18% and 14% (period A and B, respectively) of OA (Fig. 2). COA profiles (Fig. 3) reflect the expected pattern of signals related to the oxygenation of fatty acids due to cooking activities (*m/z* 29, 55, 41, 69) (He et al., 2010). Correlation between COA and HOA is not negligible, as profiles show a Pearson coefficient of $R^2 \geq 0.71$ (Table S2). However, the time series of these factors varies independently ($R^2 > 0.20$ except for Sep-Oct 2014), which allows for some confidence in their separation. The time series of COA does not show a significant seasonal trend (Fig. S12, Fig. 4). Correlation with *m/z* 55, the main COA marker, is 0.58 and 0.71 in A and B, respectively. Diel cycles present peaks at around 1 PM and 8 PM UTC, coinciding with local lunch and dinner hours and another at 9 AM probably entangled with the traffic peak at same hour (Fig. 5, Fig. S 13). Figure 5 also reveals a much higher dinner peak than the lunch peak along almost all seasons, probably related to a much thinner planet boundary layer (PBL) at 9 PM than at 1 PM.

HOA consists mainly of ions stemming from diesel exhaust from recondensed engine lubricating oil compounds (Canagaratna et al., 2010; Chirico et al., 2010). Average contributions in periods A and B are, respectively, 19% and 12% (Figure 2). HOA concentrations follow a decreasing seasonal pattern towards summer months (Fig. 4, Fig. S12) which could be associated to lower traffic intensity in August, the thicker boundary layer, the maximum recirculation induced by the highest speeds of sea-breeze and a likely faster oxidation of the primary HOA due to higher temperatures. Diel patterns (Fig. 5, Fig. S13) reproduce two traffic-associated peaks at 7 AM and 7 PM in both periods reduced during summer months, resembling traffic cycles (Fig. S14). Correlations with BC are $R^2 = 0.63, 0.68$ and with $NO_x$ are $R^2 = 0.44, 0.65$ in periods A and B, respectively, and remarkably weaker in summer periods ($R^2_{min} = 0.29, 0.51$, respectively, for BC) in accordance with a decrease in the HOA-to-BC ratio (ratio retrieved from Fig.1 and Fig. 4).

BBOA mass spectra is alike in both cold periods (Fig. 3) characterized by ions at *m/z* 29, 43, 60 and 73, fragments of anhydrosugars, such as levoglucosan, products of cellulose pyrolysis combustion reactions (Alfarra et al., 2007). BBOA is only abiding by the coldest period, from November to March (Fig. S12) and accounting for 14% and 11% of total OA in A and B, respectively. Diel cycles present the same trend in both periods (Fig. 5, Fig. S13), staying flat throughout the day and ascending from 6 PM to 0 AM, pointing to a relation to nocturnal domestic heating or breeze-driven transported pollutants from forest or agricultural fires (Reche et al., 2012) enhanced by a narrower boundary layer. Correlations with ions *m/z* 60 and *m/z* 73 are better in period B ($R^2$=0.91 and 0.61, respectively) than in period A ($R^2$=0.60 and 0.55, respectively) (Table S2).

Secondary organic aerosol (SOA) factors are resolved freely by the model unlike the constrained POA factors. They are known to be determined mostly by *m/z* 43 and *m/z* 44, which indicate the degree of oxidation (Canagaratna et al., 2015). The fractions of *m/z* 44 and *m/z* 43 over apportioned OA (f44, f43) scattered per seasons are shown in Fig. S15. In cold months dots are less concentrated and they withdraw from the expected triangle proposed by (Ng et al., 2010) due to a worst definition of SOA factors, besides some might be untrustworthy because they are close to the OA detection limit. Atmosphere is inferred to be more oxidized in period B, especially in summer months as clouds of points present lower f43 and higher f44 than in A. This is reflected in the OOA factors discussed below.

LO-OOA mass spectra differ significantly from period A to B. The *m/z* 43-to-*m/z* 44 ratio is 0.37 in period A and 1.05 in period B (Fig. 3), indicating a higher oxidation degree in period A. Contribution of LO-OOA to apportioned OA increased from period A to B although the period average is not directly comparable, as period A lacks this factor in the cold period, hence is considered as zero and leads to lower LO-OOA concentrations (Fig. 2). Considering only the three seasons excluding the Nov-Mar period, LO-OOA shows a slight decrease from period A to B (1.6 µg·m$^{-3}$ to 1.5 µg·m$^{-3}$). Seasonally, in all cases except for the already discussed summer 2014, the amount of this source decreased from period A to B (Fig. 4). In both cases, LO-OOA concentrations rise towards warmer months, which suggests SOA formation pathways might be linked to photochemical oxidation and breeze regimes. This can also be inferred by the diel patterns of LO-OOA (Fig. 5), which are flat except for a valley of around a factor of -30% from 1 to 8 PM, when temperature and irradiation reach their maximums, but also related to PBL widening and maximum sea breeze speeds at the time. The rise in LO-OOA at night could be also explained by nighttime SOA formation via NO$_3$ radical (result of anthropogenic NO$_x$ reactions) oxidation of VOC, but especially due to the land breeze prevailing during the night and transporting previously formed OA from inland areas to the coast. The valleys around 3 PM of seasonal diel cycles are more profound in April-May and June-August and the peak-to-valley ratio is also higher in period B (Fig. S13).

MO-OOA mass spectra are similar for periods A and B, even including the OOA profile from the cold subperiod in A. Excluding the Nov-March subperiod, MO-OOA shows an increase of a 90% (from 1.0 µg·m$^{-3}$ to 1.9 µg·m$^{-3}$ in A and B, respectively). The *m/z*43-to-*m/z*44 ratios are of 0.28 and 0.21, respectively, and hence, MO-OOA was less oxidized in period A (Fig. 3). MO-OOA apportions a 35% and a 42% of OA in periods A and B, respectively and this difference should be indeed widened as MO-OOA in subperiod Nov-Mar 2014-2015 is clearly overestimated (as all OOA is accounted as MO-OOA). Figure 4 supports the increase of magnitude from A to B in all seasons and highest seasonal concentrations in November-March 2014-2015 (due to a single OOA factor) followed by September-October 2014, September-October 2018 and June-August 2018. Thereof, origin of MO-OOA might be explainable from the photochemical transformation from LO-OOA oxidation, triggered amongst other causes by photochemical activity. Diel cycles show a slight increase in the middle hours of the day (Fig. 5, Fig. S13) and less pronounced seasonality in Period B rather than in A.

Concentrations of the secondary factors as a function of temperature and NO$_x$ and CO concentrations for winter (DJF) and summer (JJA) are shown in Fig. S16. A tendency of higher concentrations of SOA towards higher temperatures and NO$_x$ concentrations in winter and summer can be observed. Temperature has been reported to enhance SOA photochemical pathways from VOC; these graphs indicate, as pointed out in (Minguillón et al., 2016), that these reactions are also favored

over a certain $NO_x$ concentration threshold. However, concentrations of SOA are simultaneously high with high temperature and CO levels, inferring SOA might be enhanced not only by a highly $NO_x$-polluted ambient, but the ensemble of pollutants during severely-contaminated episodes.

$O_3$ has been reported to be an atmospheric oxidant inducing SOA production, but unexpectedly, period-averaged concentrations are substantially higher in period A than in B (Tables S4). Still, Figure S17 points out that $O_3$ is more reactive in period B, as the area of the difference between PR concentrations and those in the regional background site MSY (Montseny Natural Park, 720 m a.s.l, 40 km north–northeast of Barcelona and 25km from the Mediterranean coast) is higher during period B. This corresponds to the previously reported observation that $O_3$ diel concentrations are flatter in MSY because mountain

sites are less affected by NO titration, leading to high daily $O_3$ average concentrations (Massagué et al., 2019). Therefore, the higher the difference is between MSY and PR, the more $O_3$ has been reacting at PR. Enhanced reactivity of $O_3$ in period B could be attributed to the reaction with higher VOC concentrations resulting in the production of SOA. Also, by contrasting $O_3$ and LO-OOA diel cycles, in afternoon hours the LO-OOA increase coincides with the $O_3$ minimum, highlighting the $O_3$ capture to enhance fresh SOA.

Main contributors to high pollution episodes can be targeted monitoring the proportion of each factor to the total OA concentration as in Fig. 6 (Zhang et al., 2019). In period A, concentrations of POA decrease and then increase along with OA concentrations. However, in period B, with reduced POA concentrations respect to A, SOA represents the main pollutant even during OA growth, and more concretely, LO-OOA increases at expenses of MO-OOA. This points out that whilst POA was more responsible of high pollution episodes in A, this tendency was inverted over period B, as SOA is reasonably constant in

B through OA growth except for the largest OA class. The highest episodes of OA in B are dominated by LO-OOA, taking the lead respect to MO-OOA as its temperature-driven generation pathway is faster than the ageing to a more oxidized form. Thus, the sudden increase of primary pollutants might provoke a fast enhancement of LO-OOA, whilst MO-OOA response is slower to these stimuli.

### 3.4 Spatial origin of OA.

Figure 7 shows wind rose plots and wind-direction-dependent concentrations of OA factors. The greatest frequency (up to 15%) of winds were associated with SW wind directions and highest wind speeds are related to NE directions (Figure 7 a). Frequency of types of episodes by months are shown in Figure S4, and absolute concentrations and NR-PM$_1$ relative concentrations by episodes are shown in Figure S18.

Highest concentrations of COA are under northern and eastern winds under 4 and 8 m/s, respectively, indicative of advections

from residential areas on the north and east of the site (Fig. 7b). HOA is the most local factor, presenting its highest values with stagnation regardless of the direction despite the proximity to one of the most concurred roads of Barcelona. Hence, HOA concentrations are driven by the general variation of the urban background and not by the direct emissions from this specific road. BBOA levels are high for low wind speeds (< 6m/s) mainly coming from W and NE directions, i.e. from residential areas nearby the site. Regional production seems to be the main cause of LO-OOA as well as northern and northwestern advections,

which transport recirculating polluted air masses from the outlying industrialized areas which are canalized through the Llobregat river basin (Toll and Baldasano, 2000). MO-OOA long-range transport is deduced from the plots, since the highest concentrations are driven by stronger wind speeds from northeast in period A, and northwest, southwest and east in period B. Note that the main directions for both LO-OOA and MO-OOA in period B coincide, while the origin for LO-OOA and MO-OOA in period A seems to differ. This could be due to additional/missing source foci or changes in their advective pathways.

Regarding air masses, the proportion of OA of each source per episode is shown in Figure 8. Maximum proportion of SOA occurs under SREG episodes, coinciding with high temperatures, and coherent with the enhanced oxidation over time during these air mass recirculation episodes. High proportions of SOA is also recorded for EU and MED in period B, but for the rest

of episodes there is not a clear variation in terms of SOA contribution. Regarding absolute concentrations, the pattern is different for periods A and B (Fig. S18 a). The highest concentrations are reached during WA episodes in period A, linked to stagnation conditions, and lowest during SREG, due to major occurrence in an aforementioned anomalously cold, wet summer (Meteocat, 2020a, 2020b). For period B, the highest concentrations correspond to MED and EU episodes, while the lowest were recorded during AN, associated northern strong winds. The highest proportion of LO-OOA occurs for AW and SREG episodes, the first carrying pollutants which have crossed the whole Iberian Peninsula and the latter recirculating them along the breeze regime. MO-OOA is dominant in the highly polluted MED episodes, western advections and EU episodes, which provide long-range-driven pollutants from continental Europe. The trajectory analysis was also performed on NR-PM$_1$ species and BC, but no significant features were found regarding episodes (Fig. S18 b).

## 4.  Discussion

While bulk PM$_1$ decreased slightly from period A to B, with consistent decreases of all its main constituents except for NO$_3^-$, the average OA concentrations remain similar in both periods (4.2, 4.0 µg·m$^{-3}$). All the same, the relative contribution of the different OA sources varies significantly. On the one hand a severe reduction of POA is found from period A to B (-27%), mainly driven by a significant decrement of HOA (-37%) (Figure 2). The HOA drop is consistent with the simultaneous reduction in BC and NO$_x$ concentrations of -18% and -4% pointing to an effect of traffic-restriction implemented policies, including circulation prohibition of the most polluting vehicles during pollution episodes (enforced from December 2017) and the implementation of Euro 5 regulations. Whilst Euro 5 implies a reduction in PM emissions from diesel vehicles, the NO$_x$ emissions remain similar to previous Euro standards. This is in accordance with the larger variation of BC compared to that of NO$_x$. The difference with the even larger variation in HOA could be explained by a higher oxidation potential in the atmosphere, transforming more HOA into OOA in period B.  Diel patterns (Fig. 5, Fig. S13) reproduce two traffic-associated peaks at 7 AM and 8 PM in both periods, reduced during summer months, resembling traffic cycles (Fig. S14). The difference in the peak time from 7 AM to 8 AM for different seasons shows that the local traffic intensity is the parameter mostly driving the HOA diel variation, as both peaks correspond to 8 AM local time (due to daylight saving time), while variation in the PBL height, both starting roughly at that time, would result in less evident effect on HOA concentrations. BBOA only emerged during the coldest months (November-March) as its markers inferred, and it showed no significant variation from A to B. COA, HOA and BBOA are shown to be generated locally or short-range advected from residential areas and their relative contribution to total OA is enhanced under stagnation WA episodes, i.e., when those emissions remain in the atmosphere and no advections carry them away; but also under AN strong wind events, when despite the lower absolute concentrations (Figure S18 b), the relative contribution of primary sources increases as air is renovated and only directly-emitted pollutants arise (formation of OOA is a slower pathway). Also, the removal of SOA precursors due to strong winds could be an additional cause for the SOA proportion decrease in these scenarios.

On the other hand, SOA concentrations increased from A to B both in absolute and relative terms, representing a 45% and 60% of total OA in periods A and B, respectively. The predominance of SOA over total OA in period B remains over a 50% also for the highly polluted episodes, even with an increase of the relative POA contribution. SOA is more aged in period B, as shown by the increase of MO-OOA in detriment of LO-OOA, becoming the first the main OA constituent in B. The local production of LO-OOA or local oxidation of existing OOA seen in Figure 7, would be supported by the increase in MO-OOA concentrations during daylight for all the seasons (Figure S13), and the less marked MO-OOA local focus in A (Figure 7 b) may also evidence the lower intensity of SOA oxidation happening in A compared to B. The ratio LO-OOA-to-MO-OOA slightly decreases from A to B (mean averages of 1.1 and 1.0 (Table S4)), with lower values in autumn than summer for both A and B periods. The seasonality of LO-OOA-to-MO-OOA ratio could be partly explained by an accumulation of MO-OOA

after summer months, as temperature triggers the ageing of SOA, supported by the increase of background f44 (and conversely reduction of f43) from A to B in summer months compared to other seasons. The sensitivity to meteorological effects on the direct comparison between both summers can be avoided by comparing the LO-OOA-to-MO-OOA ratio between ordinary June and September months (avoiding anomalous and data-lacking July, August) between periods A and B. The degree of oxidation of SOA was also higher in period B (1.6 and 1.2 in June and 0.9 and 0.8 in September). The outcome of OA source apportionment in previous studies in PR compared to this one (Figure S19) shows LO-OOA-to-MO-OOA ratios of: i) 1.15 in August-September in Minguillón et al. (2016) in contrast to 1.13 and 0.68 in the August-September in period A and B, respectively; ii) 0.90 in February-March in Mohr et al. (2015) in contrast to the 0.15 of the February-March mean in period B. Even though the increasing oxidation of SOA can be observed comparing to previous works, this plot does not show a direct relation with $O_3$ concentrations. A note of caution is due here since both previous campaigns lasted less than a month, therefore comparing directly their results might be truly sensitive to meteorology or to unrepresentative events.

The enhanced SOA oxidation from A to B could be a result of the increment of the oxidation potential of the atmosphere, as reported in urban areas (Saiz-Lopez et al., 2017). Nevertheless, the $O_3$ trend, which could cause faster oxidation of SOA, is not evidently increasing contrarily to that study results. Even so, the non-increasing concentration of $O_3$ together with the increase of reactivity of $O_3$ in PR, as determined by the variations of urban and regional locations (Fig. S 17), could indicate that the concentrations in B are indeed lower owing to the higher reactivity of this pollutant. It could be argued that these results would support the hypothesis of the increase of the atmosphere oxidative potential in urban areas, but the analysis of these two one-year periods should not be extrapolated to other years. To develop a full picture of SOA evolution, additional year-long studies should be performed in PR with co-located precursors measurements such as VOC to properly assess the hypothesized oxidation of the atmosphere.

Regarding air mass transport, medium and long-range transport episodes (EU and MED) and SREG episodes show the highest proportions of SOA. The maximum LO-OOA-to-MO-OOA ratio corresponds to SREG episodes, pointing to local LO-OOA formation as the summer breeze recirculation might enhance fresh SOA. On the contrary, the lowest LO-OOA-to-MO-OOA ratio is recorded during EU and MED episodes, which advect air masses which have been travelling throughout the continent, and therefore are more aged as a result of progressive oxidation of the LO-OOA along their trajectory.

## 5. Conclusions

Characterization of non-refractory fine aerosol (NR-PM$_1$) in the urban background of Barcelona including organic aerosol (OA) source apportionment was performed regarding two nearly one-year periods between 2014 and 2018. Period-averaged PM$_1$ concentrations were 10.1 and 9.6 µg·m$^{-3}$ respectively for the so-called periods A (May 2014 - May 2015) and B (September 2017 - October 2018). Slopes between total mass concentration of Q-ACSM species and BC and PM$_1$ retrieved by a Scanning Mobility Particle Sizer were near one, but Q-ACSM overestimation caused by the use of the default relative ionization efficiency for OA, which could be lower than the actual one, cannot be discarded.

The OA was the major component in both periods under study, accounting for a 42% of total PM$_1$ in both periods. Five organic sources were identified by PMF: Cooking-like OA (COA), Hydro-Carbon-like OA (HOA), Biomass Burning OA (BBOA), Less-Oxidized Oxygenated OA (LO-OOA) and More-Oxidized Oxygenated OA (MO-OOA). BBOA was only present in the subperiod November-March and only one OOA (Oxidized OA) factor was apportioned in this cold subperiod in 2014-2015. Secondary OA (SOA, comprised by the sum of OOAs) proportion and absolute concentrations increased from the first period to the second, while primary OA (POA) concentrations were reduced. In turn, LO-OOA and MO-OOA changed its relevance in OOA contributions, being the most oxidized the promoted one from 2014-2015 towards 2017-2018. This evidences a steady

oxidation of OA components, more pronounced in warm periods, and a potential increase in atmosphere oxidative conditions resulting in higher MO-OOA contributions. Solar radiation and temperature, from noon to around 5 PM, enhance the convection processes, increasing the thickness of the planetary boundary layer and consequently causing deep valleys in factor diel cycles. Besides, photochemistry reactions have also been shown to encourage LO-OOA production and its conversion, in turn, to MO-OOA. Seasonal variation of OA contributions was also affected by air masses origin. Northern flows and stagnation episodes induced primary pollution events, although high-SOA events were driven by long-range episodes, comprising Mediterranean and European advections (mainly MO-OOA), and regional breeze-driven recirculation (mainly LO-OOA).

Average contributions of the inorganic NR-PM$_1$ were 19% and 18% for SO$_4^{2-}$, 16% and 13% for black carbon (BC), 10% and 13% for NO$_3^-$, 11% and 11% for NH$_4^+$ and <1% for Cl$^-$ for periods A and B, respectively. Hence, SO$_4^{2-}$ and BC concentrations decreased from A to B, while NO$_3^-$ ascended. Seasonal NR-PM$_1$ cycles consist of the maximization of OA and SO$_4^{2-}$ in the warmer subperiods and NO$_3^-$ (high volatility in hot conditions) and BC in the coldest. NO$_x$ were also reduced on average from A to B, as well as O$_3$ and SO$_2$. Nevertheless, O$_3$ became more reactive on period B, therefore becoming a probable promoter of OA oxidation.

To the authors' knowledge, this is one of the first times NR-PM$_1$ chemical composition and OA sources have been studied with detail allowing for interannual comparison in an urban background in the western Mediterranean basin. The results obtained highlight the role of SOA as the main source of OA and its permanency even if POA is reduced. Furthermore, the remaining unexplained gaps about the oxidation of the urban-background atmosphere of Barcelona, possible pathways of production and transformation of SOA and interfering O$_3$ processes could be the objective of further investigations.

*Data availability.* All data used in this study can be accessed here: http://dx.doi.org/10.17632/xfv7z6jzcm.1

*Author's contributions.* Study was designed by MCM, AA and XQ. Measurements were carried out by MV, MCM and CR. Data analysis was done by MV and MCM. Data interpretation was done by MV, MCM and AA, with contributions and discussions with CR and XQ. MV wrote the manuscript with contributions from MCM. All the authors revised the manuscript and agreed with its final version.

*Competing interests.* The authors declare that they have no conflict of interest.

*Acknowledgements.* This work was supported by COST Action CA16109 COLOSSAL, the Generalitat de Catalunya (AGAUR 2017 SGR41), the Spanish Ministry of Science and Innovation through CAIAC project (PID2019-108990RB-I00) and FEDER funds, through EQC2018-004598-P. Jordi Massagué is acknowledged for providing the O$_3$ data from Autonomous Governement of Catalonia. IDAEA-CSIC is a Centre of Excellence Severo Ochoa (Spanish Ministry of Science and Innovation, Project CEX2018-000794-S).

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

**Figures**

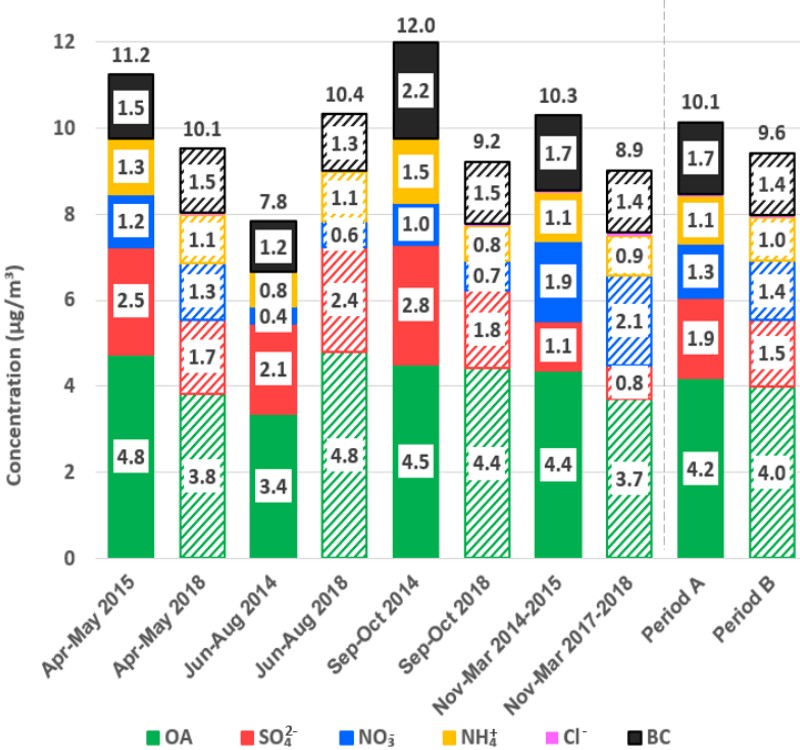

**Figure 1. Seasonal bar plot of Q-ACSM species and BC mean concentrations for periods A, with solid bars and B, with weft bars.**

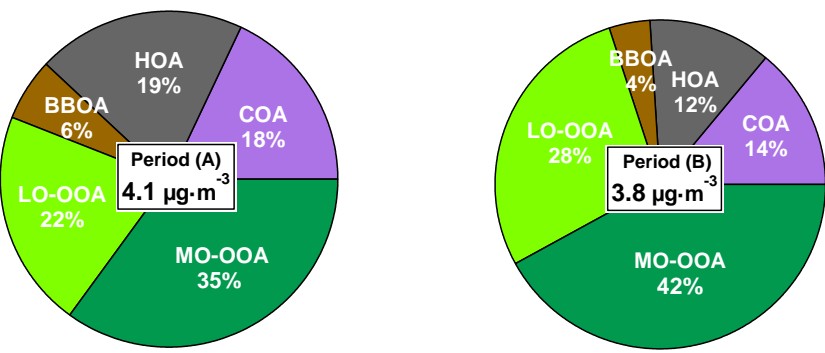

**Figure 2. Average OA source apportionment for periods A and B and concentrations of apportioned OA for period A and period B.**

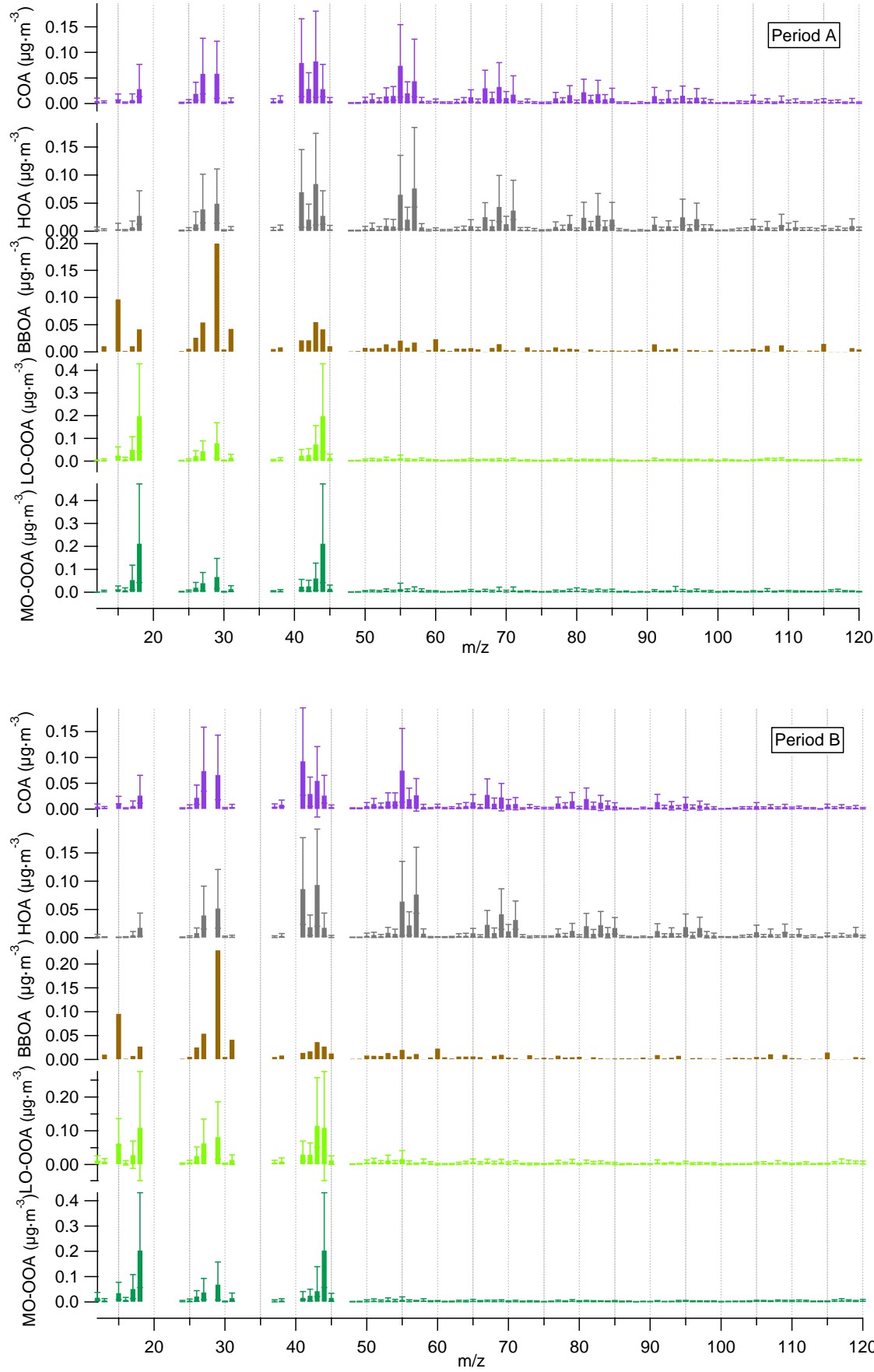

**Figure 3. Mass spectra of the five OA factors for periods A and B. Stick lines are averaged values over all seasons for a given period. Error bars represent maximum and minimum values of all seasons. Thus, BBOA is lacking error bars as it is only present in one season.**

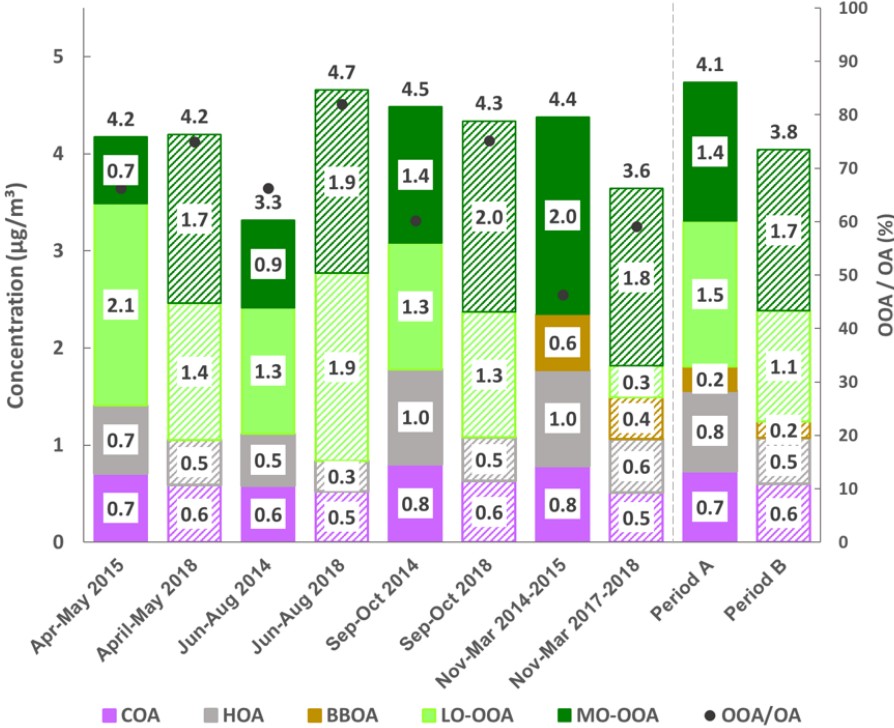

Figure 4. Seasonal OA source apportionment, with solid and weft stacked bars respectively for period A and B and OA concentrations at top. Markers show the oxygenated organic aerosol proportion with respect to total OA.

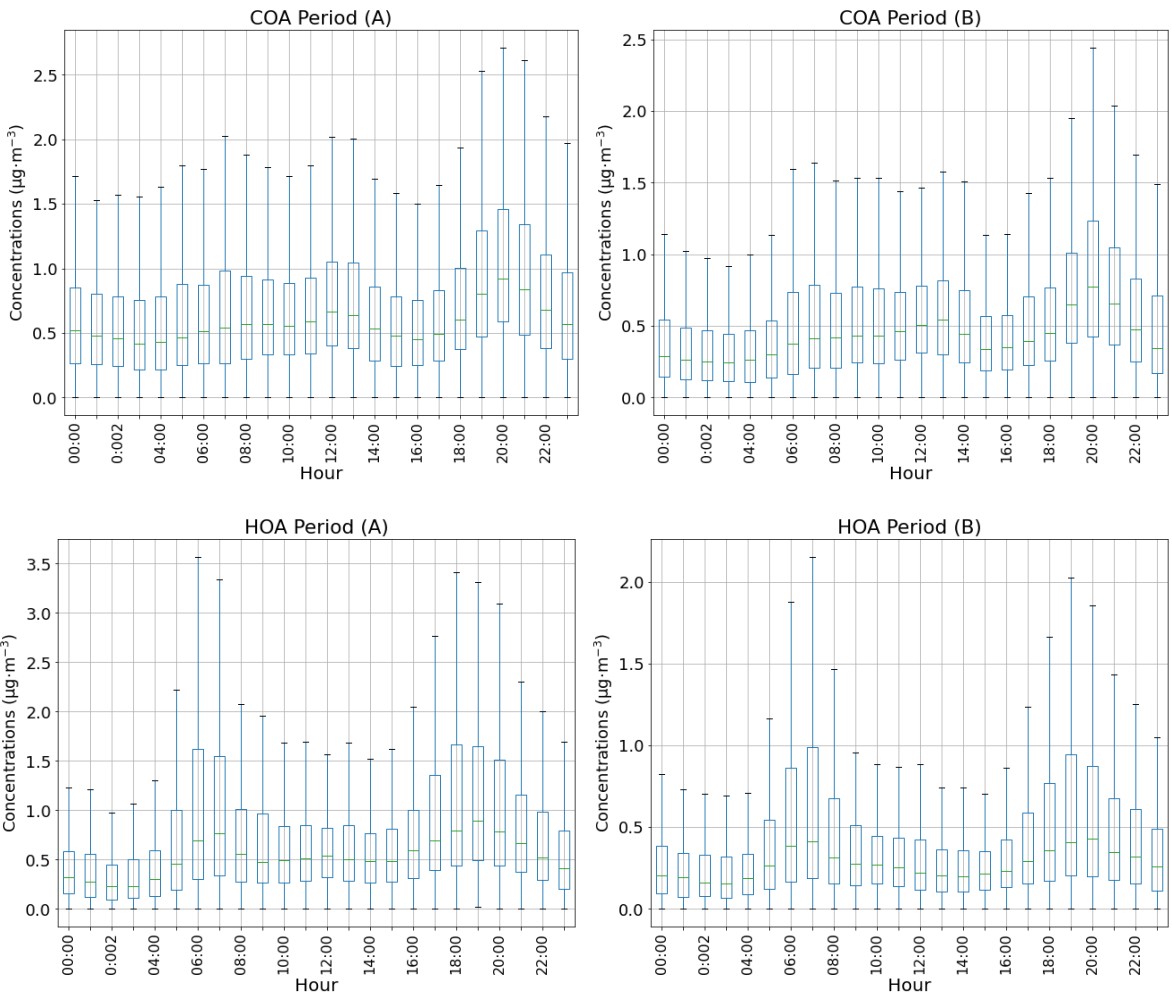

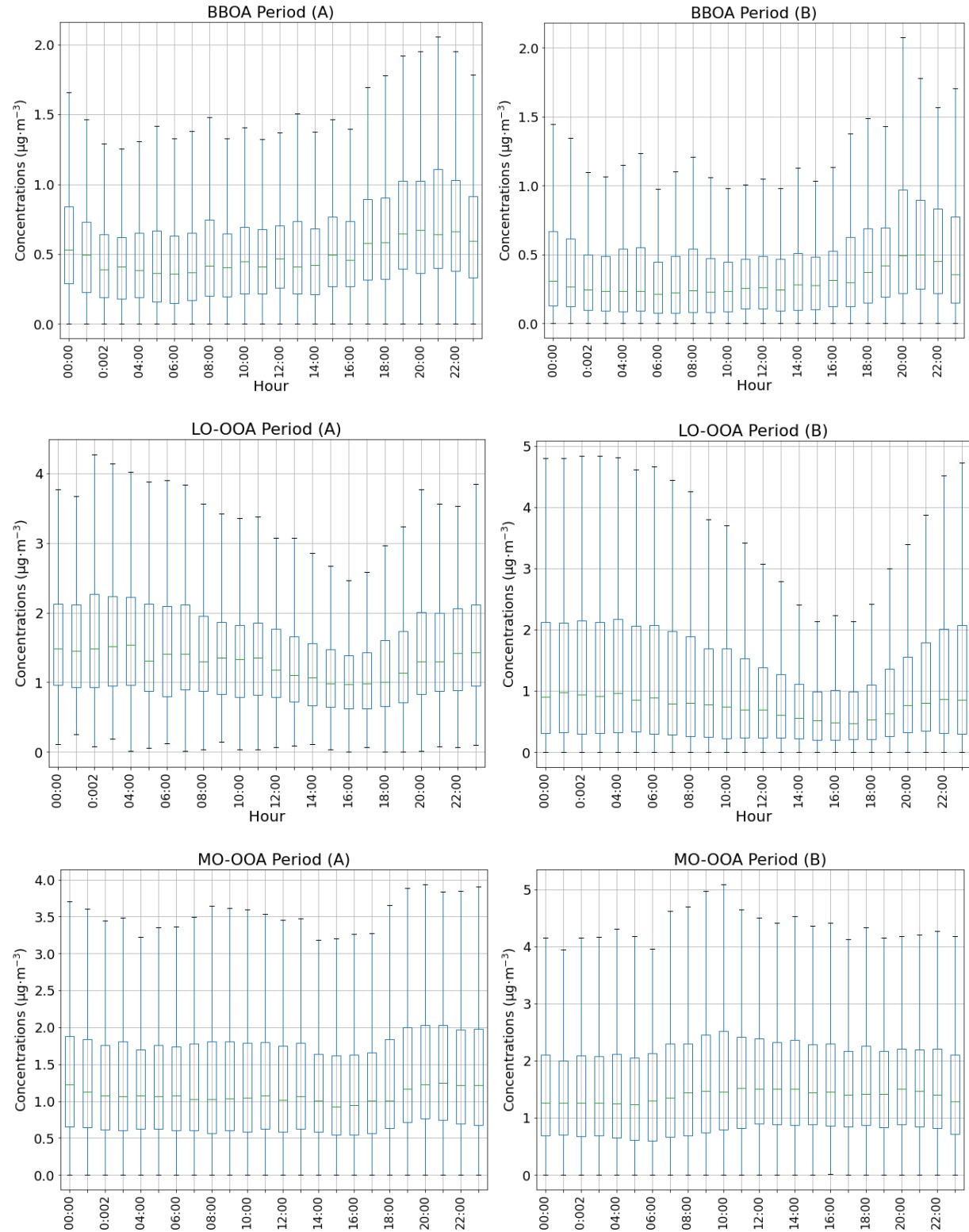

**Figure 5. Diel patterns of OA factors concentrations. Boxplots represent median and interquartile range and whiskers show maximum and minimum values. Factors are in this order COA, HOA, BBOA, LO-OOA and MO-OOA and left column corresponds to period A and right one to B.**

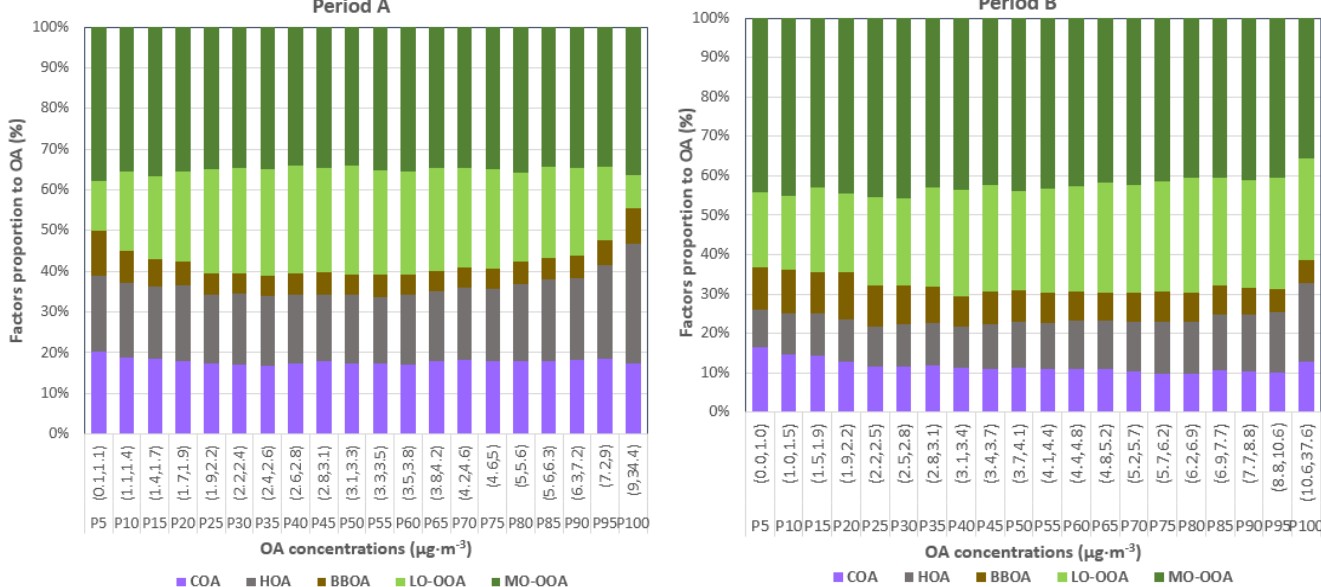

**Figure 6.** Relative contribution of organic sources as a function of OA concentrations. Ranges in brackets in the x-axis represent the range of OA concentrations for which the relative contribution is shown in each of the bins. The OA concentration bins are determined corresponding to intervals of 5 points in the percentiles, starting with P0-P5, and increasing as shown by the percentile P-x specified below the range.

**(a)**

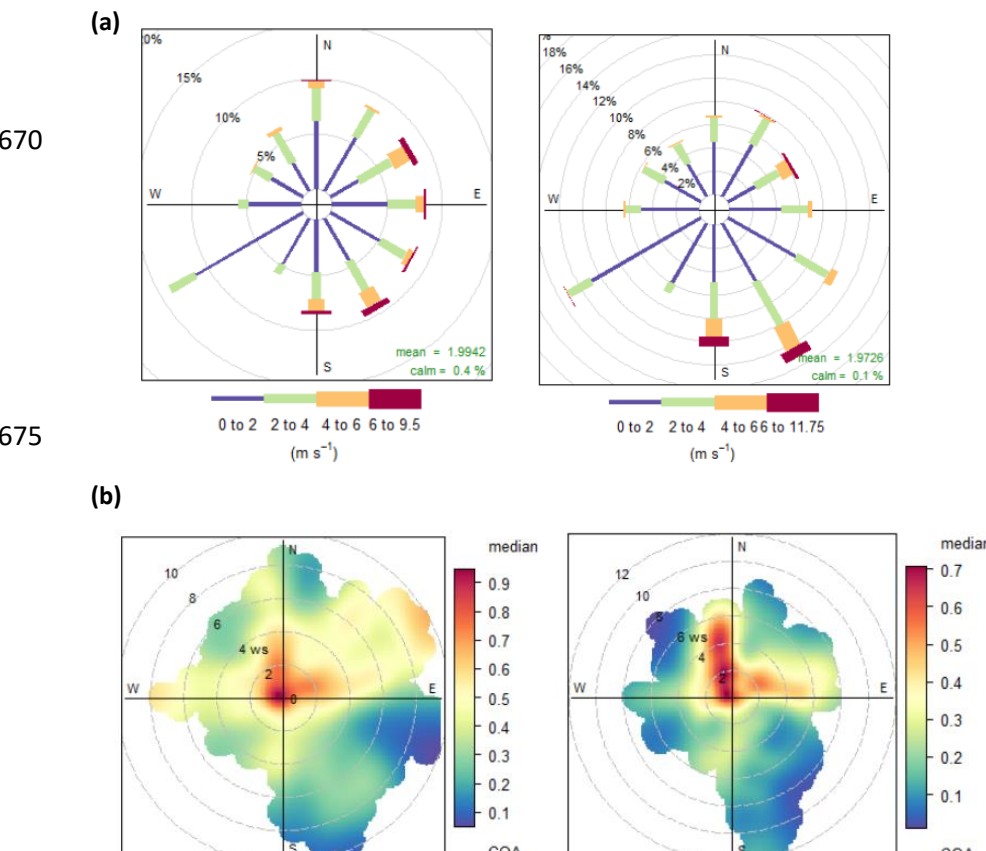

**(b)**

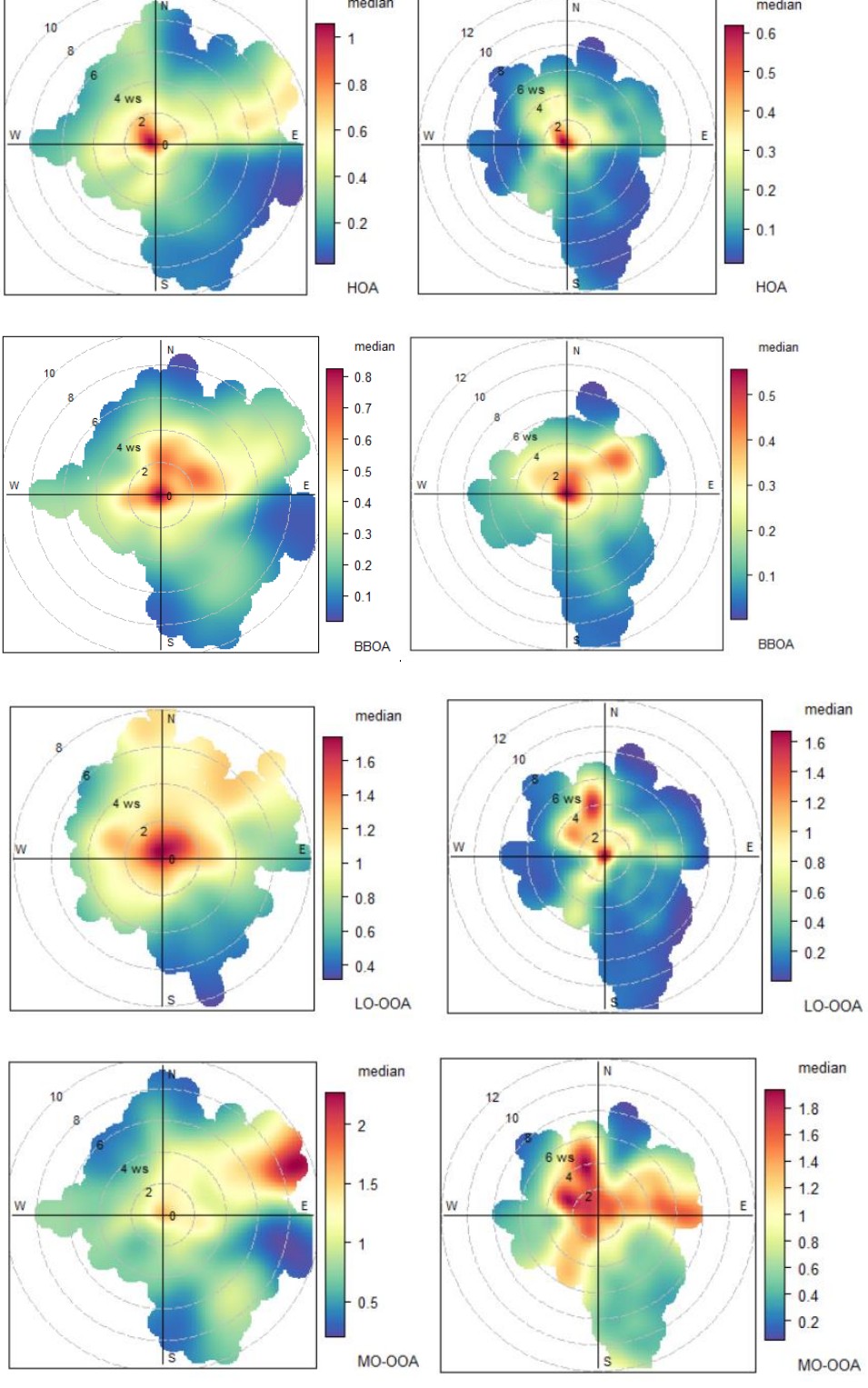

**Figure 7. Wind dependence of OA factors. (a) Windrose plots showing frequency of counts (in percentage) regarding wind direction and wind speed (m·s⁻¹). (b) Polar plots color-coded by median mass concentrations (μg·m⁻³) of OA factors listed below. In both cases, plots are arranged to the left for Period A and to the right for Period B.**

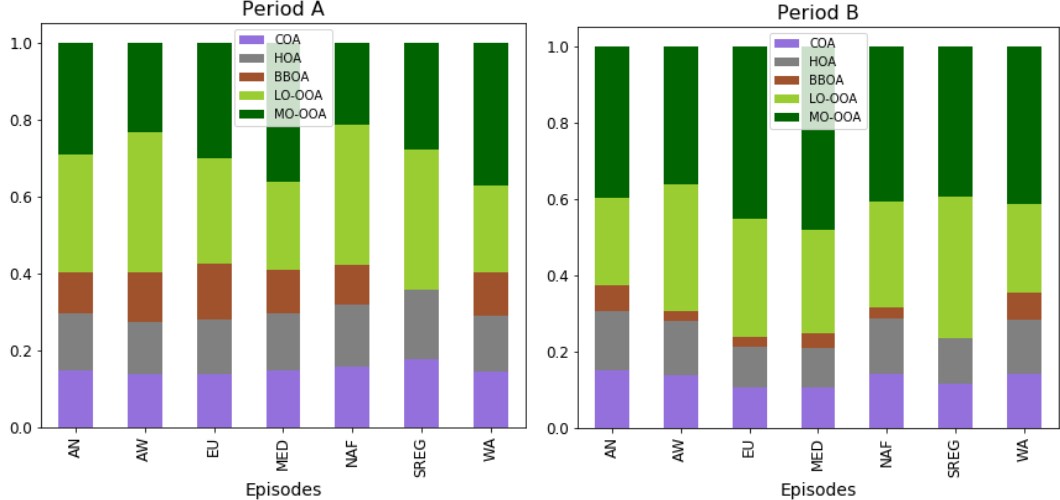

**Figure 8. Bar chart of factor proportion to total concentrations per each factor grouped by episodes for period A and B.**