# Peer review of "Interannual increase of secondary organic aerosol in an urban environment"

_Atmospheric Chemistry and Physics, 2020_

## Referee Comment (RC1) · Anonymous Referee #1 · 12 Jan 2021

The manuscript presents aerosol concentration, chemical speciation and source apportionment data for 2 one-year periods separated in time. This potentially allows for analysis of an interannual variability of aerosol sources in the urban background site of Barcelona. The secondary aerosol dominance was revealed in both periods with some differences in oxidation levels between the two. Seasonal and diurnal changes wee also discussed. The manuscript presents invaluable data and information and has a great potential to advance our knowledge in secondary aerosol formation, composition as well as transformation. While authors claim the observation of trends over the years, sufficient evidence that this is indeed a trend rather than year to year variation has not been provided in this version of the manuscript. Therefore, I recommend accepting this publication subject to major revisions listed above and below.

While the manuscript is easy to read, it would still benefit significantly from better proof-reading and improving the English language. I have indicated few points in the specific comment, but they are by far not a complete list of required language corrections. The proper tense should be used throughout the manuscript, e.g. use past tense when talking about observations for the past periods (concentrations were, not are, line 167 – 'BC are' . . . ) if referred to specific period in the past and are not recurring properties, etc.). Correct this consistently throughout the manuscript.

The main concern, however, is with the aforementioned claim of the observed trends in concentrations and atmosphere's oxidative potential, while the proof of that is missing from the current version of the manuscript. The 'reduction' claimed on line 168 and elsewhere might as well be an interannual variation due to differences in meteorology. Like LO-OOA formation in B is explained by ozone potential mostly (Lines 281-283), which can really be just interannual variability of ozone without any additional information or discussion that are lacking here.

Introduction requires more information that is relevant to this paper rather than just presenting general aerosol studies. Provide more info on MO-OOA/LO-OOA and secondary in general, discuss the atmospheres oxidative state and its changes as these are important for this manuscript. Provide the state-of-the-art that is relevant to this paper.

There are several references to other studies (line 57 and elsewhere) performed with similar instruments at similar location, like Minguillón et al., 2016 and others, but details on how this study is different from the cited ones are missing. This could be highlighted in the introduction with while the study by Minguillón et al. (2016) found such and such, they did not show this and this, therefore, in this study we . . . or something along these lines. Do it for all studies performed with an ACSM in Barcelona.

A flow of the method section can also be improved. Currently, information on PMF is scattered all over the place. E.g., if you haven't changed or modified the code, the

information on lines 104-116, related to the PMF, is redundant, just provide a citation and add more details on the PMF version that is relevant to this paper. Paatero ME-2 was not used here, so combine the information on SoFi that is provided later with the Paatero citation. Also, the citation to (Canonaco et al., 2013) for SoFi might be more appropriate? Or have both, Canonaco and Crippa (line 124).

Similarly, for the a-value approach, there is some info on lines 117-119 and then again later on lines 130-131 as well as the description of factor selection, which first appear on lines ~125, but has no appropriate details that are presented later. Be consistent and provide all related information in one place. Moreover, do not just refer to a table or figure without describing it, a reader is not supposed to make his/her own obser- vations/conclusions, this paper is about what you see from these tables and graphs (one example would be the statement that in similar forms appears several times in the manuscript: 'Differences between solutions of different number of factors for each season are shown in Table S1, Fig. S 2 and chosen seasonal profiles in Fig. S3.' – so what are these differences? Reference to Table S1 (lines132-136) does not provide details of what we are supposed to see either. Provide details, do not expect a reader to analyze the table by him/herself.

Lines 165-166: discuss all tables and graphs that are included, there is no point of providing them otherwise.

Lines 162-163: it does not seem that OM RIE is the only problem, all compounds are overestimated, can this, thus, be IE problem instead? Similarly, the statement in the conclusions (line 350) refers to OM RIE problem, but there is no discussion why other compounds are systematically overestimated? Instrument or location dependent RIE is a huge drawback for the technique, so you should be certain that this is really the case (problems with OM only and other compounds agree very well, etc.) when adding such a statement to the conclusions.

Finally, if significant environmental policies were implemented that influenced HOA

concentrations (Line 320), more information with references is required to base this statement on.

Specific comments: Line 21: ...SOA was found to be sensitive ... Line 26: ...SOA factors seem to be linked... Line 29: what do you mean by 'air-cleaning' episodes? Lines 37-38: correct the reference format for 'in Millán, 2014; Millán et al., 1997.' Lines 75-76: the difference between IEs for two periods is very large, explain why this is reasonable (major changes in the instrument or different instrument?); Line 81: provide CE ranges for the two periods; Line 88: 5-minute Lines 97-98: provide a link or a proper reference to the quality control document. It is not possible to retrieve it from the information provided. Line 131: sensitivity analysis? Line 154: I'm confused about the reference to different size ranges for the measurements in periods A and B ('...to differences in the particle size range measured...'), was that really the case? Which instrument? Or is this just a theoretical possibility that is not applicable to this study? Provide details if not. Line 157: provide slopes for specific compounds as well, not just $R^2$ Lines 176-178: rewrite this sentence in proper English Line 205: supposed to be reference to figure 6 rather than 5 here? Line 215: $R^2$ values in the brackets belong to BC or NOx? Line 216: provide information, refer to graph/table on where we can see these differences in ratios. Line 226: states that SOA was freely resolved, provide information that POA factors were constrained with specific a-values, where appropriate, when discussing primary factors then. Lines 271-272: rewrite to: Therefore, the higher the difference is between MSY and PR, the more ozone... Lines 277: I'd say decrease and then increase? Line 307: Fig 9 not S9b, maybe? Line 308: 'due to major occurrence in an aforementioned anomalously cold, wet summer' - double-check if it is really mentioned as I don't recall reading this. Also, can this be the explanation for differences between A and B rather than a consistent trend? Line 319: is this a reduction by 18 and 4%? And why the HOA shows a higher reduction? Discuss it.

Line 324: 'at the expense of', rewrite the whole sentence on lines 323-325 ('Digging into SOA composition, it is more aged in period B, as shown by the increase of MO-

OOA component at expenses of the LO-OOA reduction, becoming the main OA constituent 325 in period B.')

Lines 325-326: provide more details, increased potential due to what? Or the years were just different as you have referred before to the exceptional summer of A?

Line 337: Very strange that strong winds contributed to the accumulation of local pollution? Usually, these contribute to dilution of local emissions, not increase in concentration. Provide an explanation.

Lines 339-342: this is just repeating the results, without any further contribution to discussion. Add an appropriate discussion.

Lines 353-354: the reasons of 'BBOA was only present in the subperiod November-March and only one OOA factor was apportioned in the cold subperiod in 2014-2015' were never explained in the manuscript. This needs better discussion in discussion section, not just a statement.

Lines 357-359: do you refer to gradual increase or just year to year variation? as there is not enough evidence for the former yet.

Figure 2: I think there is a mistake in average concentrations number at the centre of the pies, table indicates 4.2 and 4 for A and B respectively? Figure 3: bars are shifted to the right in B? e.g. 55, 57, 60 and other m/zs do not appear at their marks on x-axis. Explain the lack of error bars on BBOA, do not expect a reader to guess. Rename LOOA and MOOA to be consistent with the text (LO-OOA and MO-OOA).

Figure 4: this figure can be moved to supplementary.

Figure 6: very strange representation of diurnal trends. It is quite confusing without any information provided in the caption. I suggest replotting this with only one day on the x-axis and playing with different colours or different panels to represent different years and seasons. Moreover, sharp rise in BBOA and consequent drop in LO-OOA looks very artificial. Figure s8 is so much more reasonable. Finally, COA patterns in Fig 6

and Fig S8 seem to be different, there is no such strong morning peak in S8 as in Fig 6. I would suggest replacing Figure 6 with S8. It only adds to the confusion, especially, that you do not discuss anything that is there in addition to S8. E.g., no discussion on seasonal and period differences that would refer to this figure is provided. Otherwise discuss it (different diurnal cycles between years and seasons for the same factor).

Figure 7: explain the values in the brackets.

Table S1: what is the Anchor number? Is it a-value? so should it be 0.3 not 03 for May 2014? No bold line for Jun-Aug 2014, no factor was selected?

Table S3: the correlation with SMPS looks strangely poor, atypical for such type of measurements. Double check if everything was in order (instrument performance, analysis, if there is no shift in time between the instruments, etc.).

Table S3: ACSM nitrate being larger than the offline concentration is quite strange. Usually, offline instruments have larger cut-offs, even if both are PM1, and sample higher nitrate concentrations than an ACSM. Could that be IE problem? This also relates to the comment on OM RIE, is there only OM problem or other compounds as well, like NO3, which would rather point to IE, not RIE?

Figure S10: you could get rid of points that are below detection limit for this graph, this would make it clearer.

Canonaco, F., Crippa, M., Slowik, J. G., Baltensperger, U., and Prévôt, A. S. H.: SoFi, an IGOR-based interface for the efficient use of the generalized multilinear engine (ME-2) for the source apportionment: ME-2 application to aerosol mass spectrometer data, Atmos. Meas. Tech., 6, 3649-3661, 10.5194/amt-6-3649-2013, 2013.

[Figure]

---

## Referee Comment (RC2) · Anonymous Referee #2 · 9 Feb 2021

Review

The Marta Via et al. manuscript aims to compare the aerosol chemical composition measured by a Quadrupole ACSM at the urban background station of Palau Reial (Spain) during 2 periods of one year each (May 2014 – 2015 and Sept. 2017 – Oct 2018). The organic aerosol mass concentration is further described using source apportionment analysis (ME-2) allowing the identification of 5 factors (HOA, COA, BBOA, LO-OOA, and MO-OOA). Based on these results, the authors investigate the seasonal change in the aerosol chemical composition as well as the variation between the two sampled years. Although the present work follows a classical approach regarding the analysis of the Q-ACSM dataset, the comparison of the 2 sampling periods, as well as the influence of the wind direction and trajectory analysis, represents an important

added value of the manuscript. Therefore, the present work is fully suitable for the focus of ACP. However, this work is suffering from some conceptual issues on the comparison between the two years to properly support the conclusions and major revisions are needed before a potential publication on ACP.

Major comments - The paper is focused on the change of the aerosol chemical composition (included organic aerosol sources) over 4 years. However, the dataset itself covers only the years 1 and 4 of these 4 years. So in reality, the authors compared the mass fraction and chemical composition only between two years and can therefore hardly conclude on a possible trend since no information on the variation of the aerosol chemical composition and masses between the 2 periods. The strong limitation for such an approach is to properly quantify the influence of the interannual variability of the meteorological conditions and its impact on the aerosol chemical composition. How did the authors consider this variability? Can their conclusion be bias by the fact that the summer of period A was colder and wetter than a typical summer? Consequently, it is difficult to consider it as an absolute evolution trend over the 4 years. - Another critical issue is related to the estimation of the chemical composition change between the 2 years itself. The conclusions are based on a direct comparison of the average mass concentration values obtained during periods A and B. For example, the authors did not convince me concerning a possible decrease of the organic mass concentration by 5 % between periods A and B when the respective average concentration at 4.0 (+/- 2.8) and 4.2 (+/- 2.8) $\mu$g/m$^3$ (line178) but at the same time this difference is considered as "similar" (line 317). I agree that there is a difference of 5 % between the two values but is this difference significant enough? Could it be rather considered to be inside the instrumental uncertainties or the resulting of the interannual variability? The same comments can be made for all chemical species as well as for the factor analysis. Here a summary table including all average mass concentrations of each species and factors as well as their estimated change including a proper discussion is necessary. Again, in absence of continuous measurement over the 4 years, it is difficult to conclude about a significant change without any statistical analysis for comparing the 2 years. Maybe
a seasonal approach could also be considered to affine the conclusion of the authors. Moreover, the discussion on the spatial origin of the OA is weakening the argumentation on a possible change in the SOA mass concentration between periods A and B. Line 308, the authors mentioned that the summer of period A was anomalously cold and wet. Could it be that the reported difference of SOA mass concentration between the two periods, been influence by this untypical summer leading to lower SOA formation due to bad weather conditions for period A compare to more typical summer? This is also supported in Figure 8, which shows that LO-OOA and MO-OOA were not coming from the same area during periods A and B. - Off-line filter analysis: I would expect more details regarding the off-line measurements. What was the sampling time (24h, midnight to midnight)? On the manuscript, only the ICP-AES is mentioned, while the SI referred to ICE-AES and ICP-MS. Could you please clarify? Please also provide the type of the instrument, the manufacturer, and under which condition the analysis was made? A similar comment can be made for the Ion chromatography analysis and the selective electrode. Last but not least, it is known that the value of the OC and EC strongly depend on the reference method used (Chiappini et al., 2014, Cavalli et al., 2010), it is important to provide also the reference of the instrument used for the analysis as well as the applied method (NIOSH-like, EUSAAR-2, IMPROVE_A). - The description and the discussion of the source apportionment analysis need to be improved and to be combined in a single place in the manuscript. Moreover, the authors included a lot of figures and tables on the supplementary information that are not described at all. In a more general comment, all figures and tables present in a manuscript have to be properly discussed by the authors. This is quite a pity and the description of the factor analysis will certainly gain clarity if the authors described the results of the intermediate factor solution and why they were satisfied by their final factor solution as well as factor identification. For example, the authors considered the presence of a biomass burning factor based on the monthly average of the ratio f60 and f73, what about the ratio of f44 vs f60? Could this average mask some biomass burning events? Did the authors also investigate the presence of BBOA for the other

seasons in an early stage of their source apportionment analysis? I also have a small concern regarding the averaged values of the BBOA contribution (6% for period A and 4% for period B, figure 2). When comparing the different average values (figures 2, 5, 7, and 9), I got the impression that the BBOA contribution is slightly overestimated compare to what can be expected from figure 5 (it could also be an averaging effect). This is also the case when regarding the BBOA contribution between periods A and B in figure 7, the overall BBOA mass faction seems to be always higher than 10% (especially for period B). Did the authors consider the BBOA mass concentration as 0 $\mu$g/m3 or as no value for seasons when BBOA was not identified? Also, a discussion will help to understand the choice of the single OOA factor for period A during the Nov.-March period. How the two mass spectra of this MO-OOA compared to the 3 other MO-OOA of period A? - The discussion on the special origin of the aerosol is very interesting but it would be extremely helpful to include a map showing the surrounding of the station when describing for example the presence of roads, or residential areas. Similarly, the different air mass trajectories analysis is poorly described in the current version of the manuscript. Here again, it would be helpful for the reader to know which method was used to classified the trajectory, to characterize each of them by their frequency of occurrence, their meteorological characterization (main period of occurrence, average temperature, wind speed...), and to include a figure with the different class of trajectories. Last but not least, it is damageable that the inorganics species and BC are not included in the trajectory analysis. It would certainly provide added values to support the discussion. - The statements of the authors must be systematically supported by numbers. Only mentioning "highest", "higher", "lower", "high proportion", "lowest" is hard to understand if there are no clear references.

Minor comments: - Abstract: line 13: is the different between 10.1 and 9,6 $\mu$g m-3 statistically significant? - Line 15: please support the term "significant" decrease and "higher" by numbers. - Line 21: how the authors defined SOA based on their PMF results? - Line 37: please check the uniformity of the referenced labeling. - Line 65 and rest of the manuscript: please choose between Q-ACSM or ACSM. - Line 70: 70

eV - Line 75: Could the authors please precise how often the ACSM was calibrated during each period? Are the IE and RIE a single calibration or an average value? - Line 86: Could the authors precise the range of the OPC measurements and how is it comparing to the SMPS? - Line 93: is the SIR S-5012 NOx monitor equipped or not with a blue-light converter? - Line 93-96: Were all the additional measurements performed at the same place or not? - Line 100: Could the authors refer her to section 3.1. At this stage, it is not clear that the comparison will be discussed later on. - Line 116 and follow: only ME-2 allows to constrain a factor and the use of the a-value, PMF cannot do that. - Line 125: Please also include here reference to Canonaco, F., Crippa, M., Slowik, J. G., Prévôt, A. S. H., and Baltensperger, U.: SoFi, an IGOR-based interface for the efficient use of the generalized multilinear engine (ME-2) for the source apportionment: ME-2 application to aerosol mass spectrometer data, Atmos. Meas. Tech., 6, 3649-3661, doi:10.5194/amt-6-3649-2013, 2013. - Lines 121, 128: The authors should define their acronym the first time they use them in the manuscript (here BBOA, HOA, COA), not a few lines later (here line 138-140). - Line 153: The results of the comparison between ACSM and co-located measurements will gain an understanding by also including the plots on the SI and not only a table. - Line 153 and Table S3: please check the homogeneity of the fitting parameters. The authors used either one or three digits for the same analysis. Only, 2 digits are necessary here. - Line 157: The authors should not only look at the $R^2$ to consider that the agreement is "very good" but also on the slopes and intercepts. For example, nitrate and sulfate correlation have both high $R^2$ (0.84 to 0.93, respectively) but their slopes (1.9 for the nitrate in period A and 1.32 for sulfate in period B) show a large overestimation by the ACSM. This must be discussed too before concluding on a "good agreement". - Line 160: I do not fully agree with the conclusion of the authors. The comparison between the organic mass measured by the ACSM and the OC from the filter can also be impacted by the reference method used for the OC estimation. Did the authors try to convert their organic mass into OC directly compared to the offline OC? - Line 165: why not including the time series of the ACSM measurements directly in the main

manuscript. - Line 262: Is the number of "high", not a bit too "high" in this sentence? - Line 291: reference to Fig. 8 is missing? - Line 297: could you please add a reference here? -line 319: Please refer to the corresponding figure. - Figure 2: Should the total mass of organic not be 4.2 $\mu$g/m3 for period B? - Figure 8: The legend of the color code is missing. - Figures 3,4,6, S1, S2, S3, S4, S5, S9, S12: legend of the x-axis is missing. - Figure S8: The legend of the y-axis is missing. - Diurnal figures: It is hard to identify the different seasons as well as the hours on the present figures. I would strongly suggest separating each diurnal profiles. - Supplementary section 1: please provide more details regarding the sampling regime, the analytical instrumentation, and methods used. - Tables on the supplementary information: Please reorganize the tables to have a complete table on one page.

References: Cavalli, F., Viana, M., Yttri, K. E., Genberg, J., and Putaud, J. P.: Toward a standardised thermal-optical protocol for measuring atmospheric organic and elemental carbon: the EUSAAR protocol, Atmos. Meas. Tech., 3, 79-89, DOI 10.5194/amt-3-79-2010, 2010. Chiappini, L., Verlhac, S., Aujay, R., Maenhaut, W., Putaud, J. P., Sciare, J., Jaffrezo, J. L., Liousse, C., Galy-Lacaux, C., Alleman, L. Y., Panteliadis, P., Leoz, E., and Favez, O.: Clues for a standardised thermal-optical protocol for the assessment of organic and elemental carbon within ambient air particulate matter, Atmos. Meas. Tech., 7, 1649-1661, 10.5194/amt-7-1649-2014, 2014.

---

## Author Comment (AC1) · 24 Mar 2021

**Reply to reviewers (acp-2020-1244)**

The authors would like to thank the reviewers for their comments and suggestions, which helped improving the quality of this work. A new version of the manuscript has been prepared following the suggestions from the reviewers. We provide below detailed replies to each of the comments in a point-by-point manner.

**Anonymous Reviewer #1**

1. **The manuscript presents aerosol concentration, chemical speciation and source apportionment data for 2 one-year periods separated in time. This potentially allows for analysis of an interannual variability of aerosol sources in the urban background site of Barcelona. The secondary aerosol dominance was revealed in both periods with some differences in oxidation levels between the two. Seasonal and diurnal changes were also discussed. The manuscript presents invaluable data and information and has a great potential to advance our knowledge in secondary aerosol formation, composition as well as transformation. While authors claim the observation of trends over the years, sufficient evidence that this is indeed a trend rather than year to year variation has not been provided in this version of the manuscript. Therefore, I recommend accepting this publication subject to major revisions listed above and below.**

The referees have depicted that this study should be approached as an interannual study of the organic aerosol in an urban background rather than an evolution along a 4-year period. The authors have considered this highly reasonable remark and have changed the focus of the manuscript as well as its title:

*Interannual increase of secondary organic aerosol in an urban environment.*

The discussion has been modified to acknowledge the limitation on trend conclusions based on the available data. Additional measurements from previous studies, have been provided and properly explained, in order to present more information on the evolution of air composition over time. Still, the limitations in the comparison, given that the previous studies correspond to short campaigns (one-month) is explained and considered. Please see specific responses below for more details on how this has been addressed.

2. **While the manuscript is easy to read, it would still benefit significantly from better proof- reading and improving the English language. I have indicated few points in the specific comment, but they are by far not a complete list of required language corrections. The proper tense should be used throughout the manuscript, e.g. use past tense when talking about observations for the past periods (concentrations were, not are, line 167– 'BC are'...) if referred to specific period in the past and are not recurring properties, etc.). Correct this consistently throughout the manuscript.**

The tense of this specific sentence was changed, as well as tenses in other sentences referring to specific periods in the past. Besides, a thorough revision of the English language has been performed.

3. **The main concern, however, is with the aforementioned claim of the observed trends in concentrations and atmosphere's oxidative potential, while the proof of that is missing from the current version of the manuscript. The 'reduction' claimed on line 168 and elsewhere might as well be an interannual variation due to differences in meteorology. Like LO-OOA formation in B is explained by ozone potential mostly (Lines 281-283), which can really be just interannual variability of ozone without any additional information or discussion that are lacking here.**

The authors agree that, as the referee pointed out, the previous version of this study was lacking enough demonstration to acknowledge a 4-year tendency. Both the term "reduction" and the LO-OOA relation to the oxidation potential were not supported by enough proof in the previous version of the manuscript. In this reply, the authors aimed to provide sufficient evidence to reinforce these statements if possible with the available data. To tackle this issue, previous and posterior available data has been provided in order to support or disregard the existence of such a trend.

Figure R1 shows the concentrations of $PM_1$, and OC, EC concentrations from external measurements for the 2012-2018 and 2005-2017 periods, respectively, and the measurements of $PM_1$, OA and BC for the specific periods under study in this manuscript. The seasonal Mann-Kendall test was applied to this long-term data in order to determine the existence of a trend, and in the positive case, whether this was an ascending or descending trend. Despite the seasonal Mann-Kendall test was applied to long-term data, this approach would not be suitable for the A-to-B period, since its performance requires more than four points for a meaningful interpretation, and there are only three repeated seasons at most. Hence, this test was only applied to long-term time series of variables which could be linked to the ones shown in this study (Table R1), $PM_1$ to NR-$PM_1$+BC, EC to BC and OC to OA, to ensure a meaningful trend-analysis. The seasonal Mann-Kendall statistics show decreasing trends in all the original length cases (normalized test statistics and slopes of 17.46, -0.0002 for $PM_1$, 8.02 -0.0045 for EC and 7.97, -0.0078 for OC). The gradual decline of $PM_1$ from 2012 to 2018 is consistent with the decrease found in the present study from A to B. Therefore, the reduction of a -5% could be interpreted as part of this steady drop along 6 years. EC and OC present significant decreasing trends for the period 2005-2017 (confirmed by the seasonal Mann-Kendall test), although the decrease in the concentrations from 2012-2017 is not identified from this method. Hence, the variation in concentrations found in our two campaigns A and B cannot be interpreted as a significant trend, although it cannot be completely discarded either.

[Figure]

**Figure R1. Long-term time series compared to this study two-period time series for respectively, (a) $PM_1$ from OPC and Q-ACSM + MAAP. (b) OC from filters and OM from Q-ACSM. (c) EC from filters and BC from MAAP.**

**Table R1. Mann-Kendall test statistics. The resolution of the data is hourly for PM₁ and 4 days for EC and OC.**

| Seasonal Mann-Kendall Test | | Trend | p-value | Normalized test Statistic | Slope (Theil-Sen) | Intercept |
|---|---|---|---|---|---|---|
| **PM₁ (2012-2018)** | | Decreasing | 0.0 | 17.46 | -0.0002 | 7.85 |
| **EC** | **2005-2017** | Decreasing | 1.11e-15 | 8.02 | -0.0045 | 1.11 |
| | **2012-2018** | No trend | - | - | - | - |
| **OC** | **2005-2017** | Decreasing | 1.55e-15 | 7.97 | -0.0078 | 2.39 |
| | **2005-2017** | No trend | - | - | - | - |

Figure R1 has now been included in the revised Supplementary as Figure S9 and the paragraph in section 3.2 (previous line 168) will now read as follows:

*Data overview for periods A and B is shown in Table S4 and NR-PM₁ species and BC time series in Figure S8. Average PM₁ concentrations (± standard deviation) resulting from the sum of NR-PM1 components and BC were 10.1 ± 6.7 µg·m⁻³ during campaign A and 9.6 ± 6.6 µg·m⁻³ during campaign B (variation of a 5% from A to B). A drop of a -5%, -21%, -9% and -18% is shown for of OA, SO4²⁻, NH4⁺ and BC, respectively, although NO3⁻ and Cl⁻ had a positive variation of +8% and a +20% from period A to B. Previous data for PM₁, (2012-2018) showed a decreasing trend according to Mann-Kendall test (Figure S9) and a reduction over a longer period (2005-2017) was also determined for OC, EC. These long-term reductions could imply that PM₁ and the OC and EC related pollutants (OA and BC) decreases found in this study reflect the tendency of the last years.*

Information on gaseous pollutants concentrations is also available for the period 2012-2018 in Figure R2, showing annual average concentrations of NO, $NO_2$, $NO_x$ and $O_3$. Whilst the trend for NO, $NO_2$, $NO_x$ is monotonically descending along the years, the ozone trend is not clear. It has been reported widely that these two variables are linked and anti-correlated, but their relative variation is not coupled during the latter years in Barcelona, as they do not depend exclusively on the other component but there are other reactions that can take place in the atmosphere and affect their concentrations.

[Figure]

**Figure R2. Trends of gas-phase pollutants at Palau Reial during the period 2012-2018.**

As the reviewer points out, there is not enough evidence to confirm this progressive increase of the oxidative potential of the atmosphere, as $O_3$ does not show a clear increasing trend. However, Figure S17 in the revised version (previously, Figure S12) shows the increase of ozone reactivity from period A to B due to titration enhancement in PR, implying therefore that the production of ozone in B could exceed that in A as it has enhanced its reactivity but the firstly emitted/formed amount is unknown. This does not imply an increasing tendency necessarily, but it justifies the interannual increase of the oxidation of SOA (and a consequent decrease of the LO-OOA-to-MO-OOA ratio) from period A to B, not followed by an $O_3$ increase.

In order to have a broader view of the oxidation degree evolution, information from previous campaigns has been compared according to the suggestions made in the review process. Figure R3 shows the ratio LO-OOA-to-MO-OOA over the previous works and the present study considering ozone concentrations. Data from February-March 2009 were reported in Mohr et al. (2012), and data from August-September

2013 in Minguillón et al. (2016). The data from the present study has been grouped in months selected to match the months in which the previous campaigns took place. In this graph there is not a clear relation between LO-OOA-to-MO-OOA ratio with the $O_3$ concentration, but its decrease reinforces the hypothesis of the SOA oxidation throughout these years both in summer and winter months. Note that only two data points are available here (Mohr et al., 2012 and February-March of Period B), as February-March from period A could not be calculated as only one OOA was retrieved in the November-March season. Figure R3 was included in the Supplementary (Figure S19) and commented in the discussion section as follows:

*The degree of oxidation of SOA was also higher in period B (1.6 and 1.2 in June and 0.9 and 0.8 in September). The outcome of OA source apportionment in previous studies in PR compared to this one (Figure S19) shows LO-OOA-to-MO-OOA ratios of: i) 1.15 in August-September in Minguillón et al. (2016) in contrast to 1.13 and 0.68 in the August-September in period A and B respectively; ii) 0.90 in February-March in Mohr et al. (2015) in contrast to the 0.15 of the February-March mean in period B. Even though the increasing oxidation of SOA can be observed comparing with previous works, this plot does not show a direct relation with $O_3$ concentrations. A note of caution is due here since both previous campaigns lasted less than a month, therefore comparing directly their results might be more sensitive to meteorology or to unrepresentative events.*

[Figure]

**Figure R3. Scatterplot of the LO-OOA-to-MO-OOA ratio vs. time as a function of $O_3$ concentrations (marer size) for the present and previous studies in February-March (blue markers) and August-September (red markers).**

Moreover, preliminary and unpublished 2019 Q-ACSM data in Palau Reial has been analysed and included in Figure 4, where the monthly cycle for period A, period B and January – August 2019 is presented. The f43-to-f44 ratio shows a decrease from period A to B and also from B to the January-August 2019. This ratio is related to the degree of SOA oxidation; therefore, this result would underscore the stated hypothesis of the increasing oxidation potential of the atmosphere reflected in an increasing oxidation state of the ambient OA.

[Figure]

**Figure R4. Monthly averages of f43-to-f44 ratio for periods A (May 2014 - May 2015), B (Sep 2017 – Oct 2018) and Jan-Aug 2019.**

All things considered, although irrefutable evidence has not been presented to state an oxidation trend due to the increasing oxidative potential as reported in (Saiz-Lopez et al., 2017) for urban backgrounds, significative indications that this could be the case have been shown. Prone increase of the potential oxidation of the atmosphere could be a possible explanation for the observed positive variation in the LO-OOA-to-MO-OOA ratio from period A to B, especially in summer months.

The discussion section was changed to this:

*The enhanced SOA oxidation from A to B could be a result of the increment of the oxidation potential of the atmosphere, as reported in urban areas (Saiz-Lopez et al., 2017). Nevertheless, the $O_3$ trend, which could cause faster oxidation of SOA, is not evidently increasing contrarily to that study results. Even so, the non-increasing concentration of O3 together with the increase of reactivity of $O_3$ in PR, as determined by the variations of urban and regional locations (Fig. S 17), could indicate that the concentrations in B are lower owing to the higher reactivity of this pollutant. It could be argued that these results would support the hypothesis of the increase of the atmosphere oxidative potential in urban areas, but the analysis of these two one-year periods should not be extrapolated to other years. To develop a full picture of SOA evolution, additional year-long studies should be performed in PR with co-located precursors measurements such as VOCs to properly assess the hypothesized oxidation of the atmosphere.*

4. **Introduction requires more information that is relevant to this paper rather than just presenting general aerosol studies. Provide more info on MO-OOA/LO-OOA and secondary in general, discuss the atmospheres oxidative state and its changes as these are important for this manuscript. Provide the state-of-the-art that is relevant to this paper. There are several references to other studies (line 57 and elsewhere) performed with similar instruments at similar location, like Minguillón et al. (2016) and others, but details on how this study is different from the cited ones are missing. This could be highlighted in the introduction with while the study by Minguillón et al. (2016) found such and such, they did not show this and this, therefore, in this study we... or something along these lines. Do it for all studies performed with an ACSM in Barcelona.**

Introduction was improved as it follows:

*Fine particles (PM$_1$, those with aerodynamic diameter <1 µm) have a significant impact on human health (Trippetta et al., 2016; WHO, 2016; Yang et al., 2019), climate (Shrivastava et al., 2017), and visibility (Shi et al., 2014). Organic Aerosol (OA) is the main constituent of fine aerosol in the atmosphere (Zhang et al., 2007) and it can be classified regarding its origin as primary OA (POA), consisting of directly emitted OA; or secondary OA (SOA), resulting from chemical transformation of pre-existing particles, nucleation or gas-to-particle condensation. Contributions to OA are still not fully understood due to large variability of their fingerprints, response to atmospheric dynamics and transport and evolution processes dependent on site-specific meteorological characteristics and precursors provision. Field-deployable aerosol mass spectrometers have been widely used to assess these variations (e.g. Jimenez et al. (2009)) and POA sources identification; such as hydrocarbon-like OA (HOA), biomass burning OA (BBOA) and cooking-like OA (COA). The fraction of organic mass measured at m/z 44 (f44), typically dominated by the CO2+ ion and related to oxygenation, and that at m/z 43 (f43), dominated by C2H3O+, can contribute retrieving information about ageing and oxidation state of SOA (Canagaratna et al., 2015). In Zurich, for instance, f44 during summer afternoons, when photochemical processes are most vigorous as indicated by high oxidant – OX ($O_3$ + NO$_2$), was found similar or lower than f44 on days with low -OX, while f43 (less oxidized fragment) tended to increase (Canonaco et al., 2015). The SOA is often divided into two factors: less-oxidized oxygenated OA (LO-OOA) and more-oxidized oxygenated OA (MO-OOA).*

[revised manuscript text omitted]

5. **A flow of the method section can also be improved. Currently, information on PMF is scattered all over the place, e.g., if you haven't changed or modified the code, the information on lines 104-116, related to the PMF, is redundant, just provide a citation and add more details on the PMF version that is relevant to this paper. Paatero ME-2 was not used here, so combine the information on SoFi that is provided later with the Paatero citation. Also, the citation to (Canonaco et al., 2013) for SoFi might be more appropriate? Or have both, Canonaco and Crippa (line 124).**

The PMF general description has been shortened slightly. Equations have been kept, as they help to explain the results and, while most readers may know PMF already quite well, the authors prefer to keep these general equations for the sake of clarity (text in previous lines 107-116). The information on a-value in lines 117-119 has also been edited as suggested by the reviewer later. The citation of Canonaco et al. (2013) was added when mentioning SoFi, and the citation of Paatero et al. (1999) is kept when mentioning ME-2, which is the solver used in the SoFi tool. The description on the periods used for OA source apportionment, with the criteria used to do that, has also been edited. It is now the second paragraph in section 2.4. The former section 2.4 on PMF has been rephrased. Please see response to Comment #6 referee #1) for the completely new text of section 2.4.

6. Similarly, for the a-value approach, there is some info on lines 117-119 and then again later on lines 130-131 as well as the description of factor selection, which first appear on lines 125, but has no appropriate details that are presented later.  Be consistent and provide all related information in one place. Moreover, do not just refer to a table or figure without describing it, a reader is not supposed to make his/her own observations/conclusions, this paper is about what you see from these tables and graphs (one example would be the statement that in similar forms appears several times in the manuscript: 'Differences between solutions of different number of factors for each season are shown in Table S1, Fig.  S 2 and chosen seasonal profiles in Fig.  S3.' - so what are these differences? Reference to Table S1 (lines132-136) does not provide details of what we are supposed to see either. Provide details, do not expect a reader to analyze the table by him/herself.

An a-value introduction was added. The text was enlarged and further discussion included of Table S1 and Figure S3 in the revised version (previous Figure S2). Now, it includes discussion on why mainly all runs chosen optimize the numbers in Table S1 and why exceptions have been made for some specific runs. The new section 2.4, including modifications required in questions 5 and 6, reads as follows:

[revised manuscript text omitted]

7. **Lines 165-166: discuss all tables and graphs that are included, there is no point of providing them otherwise.**

Table S4 (previously also, Table S4) and Figure S8 (previously Figure S4) were already commented on the section of seasonal variation. Time series of co-located gases and meteorological variables are now discussed in section 3.2 in the revised version as follows:

*Time series of co-located gases (Fig. S6) show rises of SO2 in summer months likely related to shipping activity increase, O3 increases during summer linked to higher photochemical enhancement and an increase of CO during cold periods, associated to shallower boundary layers. NO and $NO_2$ show similar behaviors of reduction towards warm months and sudden increases in cold periods.*

*There are evident differences in meteorological variables from one period to the other (Fig. S7). In summer A, temperature (24.4º C) and solar radiation (259 W) are lower and average relative humidity (71%) and wind speed (2.0 m/s) are higher than in period B (27.0º C, 280W, 70.0%,1.7%), indicating a probably rainier or cloudier summer in period A.*

*Data overview for periods A and B is shown in Table S4 and NR-$PM_1$ species and BC time series in Figure S8. Average $PM_1$ concentrations (± standard deviation) resulting from the sum of NR-$PM_1$ components and BC were 10.1 ± 6.7 µg·m⁻³ during campaign A and 9.6 ± 6.6 µg·m⁻³ during campaign B (variation of a 5% from A to B). A drop of a -5%, -21%, -9% and -18% is shown for of OA, $SO_4^{2-}$, $NH_4^+$ and BC, respectively, although $NO_3^-$ and $Cl^-$ had a positive variation of +8% and a +20% from period A to B. Previous data for $PM_1$, (2012-2018) showed a decreasing trend according to Mann-Kendall test (Figure S 9) and a reduction over a longer period (2005-2017) was also determined for OC, EC. These long-term reductions could imply that $PM_1$ and the OC and EC related pollutants (OA and BC) decreases found in this study reflect the tendency of the last years.*

8. **Lines 162-163: it does not seem that OM RIE is the only problem, all compounds are overestimated, can this, thus, be IE problem instead? Similarly, the statement in the conclusions (line 350) refers to OM RIE problem, but there is no discussion why other compounds are systematically overestimated? Instrument or location dependent RIE is a huge drawback for the technique, so you should be certain that this is really the case (problems with OM only and other compounds agree very well, etc.) when adding such a statement to the conclusions.**

Not all compounds are overestimated after the correction performed on Figure S5 b and Table S3. For instance, $SO_4^{2-}$ Q-ACSM vs. off-line is 1.05. Overestimation of $NO_3^-$ and consequently also the overestimation of $NH_4^+$ is expounded in the reply to Specific Comment #44 (Referee #1), please see the

explanation there. Therefore, only the OA overestimation remains to be explained, which could be at least partially attributed to the OA RIE. Moreover, the correlation of NR-PM$_1$ (from ACSM) + BC (from MAAP) is not systematically over one, therefore, the effect of a misleading IE and consequent general overestimation should be ruled out.

9. **Finally, if significant environmental policies were implemented that influenced HOA concentrations (Line 320), more information with references is required to base this statement on.**

Period A shows higher traffic intensity than period B (Figure S14, in the previous version, Figure S9) except for the first four months of the year and August, when intensity drops significantly. This is consistent with the decrease of HOA from period A to B following the traffic trend. The causes of the traffic reduction might be linked to implemented restriction policies, e.g. prohibition of circulation in pollution episodes for the most polluting vehicles enforced in December 2017. The revised text reads as follows:

*The HOA drop is consistent with the simultaneous reduction in BC and NOx concentrations of -18% and -4% pointing to an effect of traffic-restriction implemented policies, including circulation prohibition of the most polluting vehicles during pollution episodes (enforced from December 2017) and the implementation of Euro 5 regulations.*

**Specific comments:**

10. **Line 21: . . .SOA was found to be sensitive . . .**

This change was implemented as follows:

*SOA was enhanced by a NO$_x$-polluted ambient*

11. **Line 26: . . .SOA factors seem to be linked. . .**

This change was implemented as follows:

*Both SOA factors were boosted with long and medium-range circulations, …*

12. **Line 29: what do you mean by 'air-cleaning' episodes?**

Air-cleaning episodes consist of episodes of high wind speeds which remove all pre-existent pollutants and cause low concentrations afterwards. These are likely associated to Atlantic northern or north-western advections (Pey et al., 2010). 'Air renewal episodes' could be a more accurate expression.

13. **Lines 37-38: correct the reference format for 'in Millán, 2014; Millán et al., 1997.'**

These citations have been corrected to:

*in Millán (2014); Millán et al. (1997).*

14. **Lines 75-76: the difference between IEs for two periods is very large, explain why this is reasonable (major changes in the instrument or different instrument?).**

The Q-ACSM detector was changed on September 2017, before the start of the second campaign. The IE applied was that one retrieved from posterior calibration. The following sentence was included in the latest version of the manuscript.

*The significant difference between IE values is a consequence of the change of the detector before starting campaign B.*

**15. Line 81: provide CE ranges for the two periods.**

The mean and standard deviation values are now presented in the text:

*The composition-dependent collection efficiency (CE) (Middlebrook et al., 2012) correction was applied (minimum, maximum) for Period A and B respectively: $(0.45\pm0.68)$, $(0.50, 0.99)$, exceeding CE=0.6 a 0.13% and a 1.5% of data).*

**16. Line 88: 5-minute**

This change was implemented in the manuscript.

**17. Lines 97-98: provide a link or a proper reference to the quality control document. It is not possible to retrieve it from the information provided.**

The COLOSSAL link has now been included with an (in prep.) note.

**18. Line 131: sensitivity analysis?**

Referring to the systematic a-value space exploration. Now this paragraph has been changed and this expression does not appear and it is properly explained:

*All combinations of a-values within a range of 0 to 0.5, with steps of 0.1, were explored for each set of runs with the same number of factors, in order to select the most environmentally reasonable solutions.*

**19. Line 154: I'm confused about the reference to different size ranges for the measurements in periods A and B ('. . .to differences in the particle size range measured. . .'), was that really the case? Which instrument? Or is this just a theoretical possibility that is not applicable to this study? Provide details if not.**

The sentence is mostly referring to SMPS measurements, (Table S3 and Figure S5 a in the revised version). Measurements of particles of diameters under 20 nm were discarded due to malfunctioning of the SMPS and upper particle diameter was set at 478.3 nm, a threshold widely covered by the ACSM. Hence, there is a difference between diameter range from SMPS and that from Q-ACSM. This supports the hypothesis that the slope over 1, mainly in Period B in their intercomparison is not only due to an ACSM overestimation. This explanation has been included in the manuscript as follows:

*The slopes are near one, especially in Period A. Nevertheless, SMPS underestimation of $PM_1$ could be partially attributed to differences in the particle size range measured, shifted to lower diameters for the SMPS (20 - 478.3). Moreover, overestimation of primary OA sources by Q-ACSM by a factor of 1.2 to 1.5 has also been reported (Reyes-Villegas et al., 2018; Xu et al., 2018) as a consequence of source-unspecific OA RIE application, leading in turn to $PM_1$ overestimation by Q-ACSM.*

**20. Line 157: provide slopes for specific compounds as well, not just $R^2$.**

The slopes of all compounds as well as their scatterplot have been included in the latest versions of the manuscript and Supplementary, respectively. The text has been improved as follows:

*The correlation coefficient for NR-$PM_1$ species off-line analysis of $SO_4^{2-}$, $NO_3^-$ and $NH_4^+$ is always above $R^2>0.71$, and also between OA and organic carbon (OC) ($R^2=0.73$ in A and $R^2=0.86$ in B) (Table S3, Figure S5 b). Slopes of the linear regression between Q-ACSM and off-line $SO_4^{2-}$, $NO_3^-$ and $NH_4^+$ are respectively 1.24,*

*1.90 and 1.73 for period A and 1.05, 1.76 and 1.68 for period B. The slopes of $NO_3^-$ are largely above 1, likely owing to volatilization artefacts in filters. The reason of the high slope for $NH_4^+$ is not clear, so neither determination problems in any of the filter species nor Q-ACSM overestimation cannot be completely discarded. The anion $Cl^-$ is not considered due to the very low concentrations and potential determination problems (Tobler et al., 2020).*

*OA-to-OC ratio is estimated from the slope in the scatterplot between these two variables, resulting in values of 2.69 and 2.94 in periods A and B, respectively, a 68% and 84% higher than the 1.6 value calculated with an AMS in Barcelona in the DAURE campaign (Minguillón et al., 2011; Mohr et al., 2015). Although the OM-to-OC ratio might have increased over the years since March 2009, which would be in accordance to an increasing SOA proportion found in these campaigns (2014-2015 and 2017-2018) respect to DAURE. The values found in the present study are too high, as the expected OOA values according both to chamber and ambient experiments should be around 2.3 (Canagaratna et al., 2015). This too high OA-to-OC ratios would point to filter artefacts caused by evaporation of semi-volatile OC compounds as well as similarly to the previous findings at Montsec (Ripoll et al., 2015) and Montseny (Minguillón et al., 2015), the aforementioned unspecific-source OA RIE overestimation in Q-ACSM. Even so, a steady increase of OOA oxidization over the years could be inferred due to the growth of this ratio from 2014-2015 to 2017-2018.*

**21. Lines 176-178: Rewrite this sentence in proper English**

This sentence was rephrased as follows:

*Seasonally-averaged concentrations and time series of $PM_1$ and its components for both are displayed in Figure 1, and Figure S8. In period A, the highest concentrations of bulk $PM_1$ occurred in Sep-Oct (12.0 µg·m-3) and the lowest were registered in summer (7.8 µg·m-3). Contrastingly, in period B, submicron aerosol was maximum in summer (10.4 µg·m-3) and minimum in winter (8.9 µg·m-3). The differences in occurrence frequency of atmospheric episodes and data availability might be a direct cause of these mismatches of seasonal trends.*

**22. Line 205: supposed to be reference to figure 6 rather than 5 here?**

The reference was changed to coincide with the updated version numeration of figures.

**23. Line 215: R2 values in the brackets belong to BC or $NO_x$?**

The sentence was rephrased for proper understanding:

*$R^2min=0.29, 0.51$, respectively, for BC).*

**24. Line 216: provide information, refer to graph/table on where we can see these differences in ratios.**

These ratios can be retrieved from the information presented in Fig. 1 and Fig. 4. References to these figures have been included in the revised manuscript.

**25. Line 226: states that SOA was freely resolved, provide information that POA factors were constrained with specific a-values, where appropriate, when discussing primary factors then.**

This information was included in the method sections (Section 2.4) and also in the results section 3.3 for the sake of clarity. This change was implemented in the manuscript:

In 2.4:

*The solutions space was explored for HOA and COA anchor profiles from Mohr et al. (2012) and Crippa et al. (2013) whilst OOA factors were let be resolved freely. The criteria to choose the anchor profiles consisted of*

*comparing correlations with external tracers or source markers: BC and NOₓ for HOA, m/z55 for COA, m/z60, m/z73 for BBOA, SO₄²⁻ for MO-OOA and NO₃⁻ for LO-OOA (Table S2). This led to the selection of the COA, HOA anchor profiles from Crippa et al. (2013) and the BBOA anchor from Ng et al., 2010. All combinations of a-values within a range of 0 to 0.5, with steps of 0.1, were explored for each set of runs with the same number of factors, in order to select the most environmentally reasonable solutions.*

In 3.3:

*The chosen output of PMF revealed 5 factors, interpreted as COA, HOA, BBOA, considered primary OA (POA), and two secondary organic aerosols (SOA) differentiated by their degree of oxidation in Less Oxidized (LO-OOA) and More Oxidized (MO-OOA) Oxygenated Organic Aerosol (OOA) (Fig. S 10). This is consistent with previous campaigns performed in Barcelona using both Q-ACSM (Minguillón et al., 2016) and AMS (Mohr et al., 2012, 2015) with same OA sources identified. Just for the cold season in period A, a single OOA factor was extracted. The reason of this exception is that when 5 factors were considered, in the f44 vs. f43 plot, the OOA profile f44, f43 dots were very close, inducing that the proportion of these ions was too similar and consequently that that pair of factors were a result of an OOA split. Supporting the solutions chosen, mean scaled residuals (Fig. S12) are sharply centered to zero in both cases although histogram A shows higher spread and skewness, meaning worse match between OA measured and the sum of OA factors concentrations. A-values finally employed are shown in Table S2 and time series for OA factors in Figure S12.*

**26. Lines 271-272: rewrite to: Therefore, the higher the difference is between MSY and PR, the more ozone. . .**

The sentence was rephrased to:

*Therefore, the higher the difference is between MSY and PR, the more ozone has been reacting at PR.*

**27. Lines 277: I'd say decrease and then increase?**

This change was implemented in the manuscript.

**28. Line 307: Fig 9 not S9b, maybe?**

This reference has been corrected in the manuscript.

**29. Line 308: 'due to major occurrence in an aforementioned anomalously cold, wet summer' - double-check if it is really mentioned as I don't recall reading this. Also, can this be the explanation for differences between A and B rather than a consistent trend?**

It has now been explained when presenting Figure S4 and also in the reply to Comment #3 (Referee #1) of the present document.

**30. Line 319: is this a reduction by 18 and 4%? And why the HOA shows a higher reduction? Discuss it.**

These percentages correspond to the reduction of BC and NOₓ concentrations from A to B. The Euro 5 standards imply a reduction in PM emissions from diesel vehicles while the NOₓ emissions did not vary significantly. Hence, the larger reduction in BC with respect to that of NOₓ is in accordance with the emissions variation. Moreover, NOₓ interact in the atmosphere with other pollutants, while BC is a primary pollutant that does not change or react once emitted to the atmosphere. The larger reduction of HOA with respect to BC may respond to the higher oxidation potential of the atmosphere, leading to a larger conversion of HOA to OOA. The revised text has been changed as follows:

*The severe reduction of POA found from period A to B (-31%) is mainly driven by a significant decrement of HOA (-40%). The simultaneous reduction in BC and NOₓ concentrations of -18% and -4% point to an effect of*

*traffic-restriction implemented policies, including circulation prohibition of the most polluting vehicles during pollution episodes enforced from December 2017 and the implementation of Euro 5 regulations. While Euro 5 implies a reduction in PM emissions from diesel vehicles, the $NO_x$ emissions remain similar to previous Euro standards. This is in accordance with the larger variation of BC compared to that of $NO_x$. The difference with the even larger variation in HOA can be explained by a higher oxidation potential in the atmosphere, transforming more HOA into OOA in period B.*

**31. Line 324: 'at the expense of', rewrite the whole sentence on lines 323-325 ('Digging into SOA composition, it is more aged in period B, as shown by the increase of MO- OOA component at expenses of the LO-OOA reduction, becoming the main OA constituent in period B.')**

This sentence was rephrased in the latest version of the manuscript to:

*SOA is more aged in period B, as shown by the increase of MO-OOA contribution in detriment of LO-OOA, hence the MO-OOA becoming the main OA constituent in period B.*

**32. Lines 325-326: provide more details, increased potential due to what? Or the years were just different as you have referred before to the exceptional summer of A?**

The increasing oxidation potential of the atmosphere should be considered a potential cause explaining these findings, given that despite the ozone concentrations do not show this variation, there are supporting facts of oxidation potential such as the MSY-BCN $O_3$ differences (Figure S17 in the latest version, previously Figure S12). Further detail about this oxidation potential would require additional measurements, such as $OH^-$ measurements, which we lack for the study periods.

**33. Line 337: Very strange that strong winds contributed to the accumulation of local pollution? Usually, these contribute to dilution of local emissions, not increase in concentration. Provide an explanation.**

It is true that strong winds (AN in this case) cause a decrease in all kinds of pollutants (see Fig S 18 a, S13 in the previous version of the Supplementary). However, the remaining pollutants after these episodes have been apportioned as POA likely due to the slower pathway of SOA formation and the removal of its precursors by strong winds. In Figure 8 (previously Figure 9) it is seen how despite absolute concentrations decrease in this kind of episodes, POA relative contribution increases. A more complete explanation was included in the discussion section:

*COA, HOA and BBOA are shown to be generated locally or short-range advected from residential areas and their relative contribution to total OA is enhanced under stagnation WA episodes, i.e., when those emissions remain in the atmosphere and no advections carry them away; but also under AN strong wind events, when despite the absolute concentrations being lower (Figure S18 b), the relative contribution of primary sources increases as air is renovated and only directly-emitted pollutants arise (formation of OOA is a slower pathway). Also, the removal of SOA precursors due to strong winds could be an additional cause for the SOA proportion decrease in these scenarios.*

**34. Lines 339-342: this is just repeating the results, without any further contribution to discussion. Add an appropriate discussion.**

Discussion has been extended as follows:

*SOA is more aged in period B, as shown by the increase of MO-OOA in detriment of LO-OOA, becoming the main OA constituent in B. The local production of LO-OOA or local oxidation of existing OOA seen in Figure 7, would be supported by the increase in MO-OOA concentrations during daylight for all the seasons (Figure S13), and the less marked MO-OOA local focus in A (Figure 7 b) may also evidence the lower intensity of SOA*

*oxidation happening in A compared to B The ratio LO-OOA-to-MO-OOA slightly decreases from A to B (mean averages of 1.1 and 1.0 (Table S4)), with lower values in autumn than summer for both A and B periods.*

**35. Lines 353-354: the reasons of 'BBOA was only present in the subperiod November-March and only one OOA factor was apportioned in the cold subperiod in 2014-2015' were never explained in the manuscript. This needs better discussion in discussion section, not just a statement.**

It has been explained during the revised methodology section now. Please see the detailed response to Comment #6 (Referee #1).

**36. Lines 357-359: do you refer to gradual increase or just year to year variation? as there is not enough evidence for the former yet.**

This appreciation has been explained in the reply to Comment #3 (Referee #1) and has been integrated in the manuscript conclusions.

**37. Figure 2: I think there is a mistake in average concentrations number at the centre of the pies, table indicates 4.2 and 4 for A and B respectively?**

The concentrations in the centre of the pie correspond to the apportioned OA, whilst Table S4 presents the measured OA concentrations. The difference between these two values is stemming from the PMF model over/under estimating the introduced OA input.

**38. Figure 3: bars are shifted to the right in B? e.g. 55, 57, 60 and other m/zs do not appear at their marks on x-axis. Explain the lack of error bars on BBOA, do not expect a reader to guess. Rename LOOA and MOOA to be consistent with the text (LO-OOA and MO-OOA).**

This graph has been improved fixing all the changes mentioned. The caption also includes now the explanation for the lack of error bars for BBOA:

*Error bars represent maximum and minimum values of all seasons. Thus, BBOA is lacking error bars as it is only present in one season.*

**39. Figure 4: this figure can be moved to supplementary.**

This figure was moved to the supplementary section.

**40. Figure 6: very strange representation of diurnal trends. It is quite confusing without any information provided in the caption. I suggest replotting this with only one day on the x-axis and playing with different colours or different panels to represent different years and seasons. Moreover, sharp rise in BBOA and consequent drop in LO-OOA looks very artificial. Figure s8 is so much more reasonable. Finally, COA patterns in Fig 6 and Fig S8 seem to be different, there is no such strong morning peak in S8 as in Fig6. I would suggest replacing Figure 6 with S8. It only adds to the confusion, especially, that you do not discuss anything that is there in addition to S8. E.g., no discussion on seasonal and period differences that would refer to this figure is provided. Otherwise discuss it (different diurnal cycles between years and seasons for the same factor).**

A new figure is provided for the diel seasonal plots, hopefully clearer to the reader. It has been swapped with previous Figure S8 so that the boxplots of diel cycles are now in the main text (Figure 5, in the previous version, Figure S8) and seasonal diel cycles in the Supplementary (Figure S13). Some discussion has been also added:

*Diel patterns (Fig. 5, Fig. S13) reproduce two traffic-associated peaks at 7 AM and 7 PM in both periods, reduced during summer months, resembling traffic cycles (Fig. S14). The difference in the peak time from 7 AM to 8 AM for different seasons (Fig. S13) shows that the local traffic intensity is the parameter mostly driving the HOA variation, as both peaks correspond to 8 AM local time (due to daylight saving time), while variation in the planetary boundary layer height, which start roughly at that time, and the change in wind direction and speed, also happening roughly at that time, would result in less evident effect on HOA concentrations.*

**41. Figure 7: explain the values in the brackets.**

The caption of Figure 7 now contains the info that was required for proper understanding:

*Figure 1. Relative contribution of organic sources as a function of OA concentrations. Ranges in brackets in the x-axis represent the range of OA concentrations for which the relative contribution is shown in each of the bins. The OA concentration bins are determined corresponding to intervals of 5 points in the percentiles, starting with P0-P5, and increasing as shown by the percentile P-x specified below the range.*

**42. Table S1: what is the Anchor number? Is it a-value? So, should it be 0.3 not 03 for May 2014? No bold line for Jun-Aug 2014, no factor was selected?**

'Anchor value' was referring to the a-value for the given anchor profile. It should be 0.3 as pointed. These and other required modifications have been implemented in the revised Table S1.

**43. Table S3: the correlation with SMPS looks strangely poor, atypical for such type of measurements. Double check if everything was in order (instrument performance, analysis, if there is no shift in time between the instruments, etc.).**

In this version of the manuscript, the intercomparison with OPC was removed due to the acknowledgement of season-dependent slopes, but description of SMPS results was provided instead. Also reconstructed $PM_1$ intercomparison was removed as it was used for correction of the OPC measurements and could be the source of error. Both SMPS and Q-ACSM data have been double-checked several times and no shifts or dysfunctional periods were detected. Nevertheless, it should not be discarded any malfunctioning of SMPS during period A, which would be coherent with the bad correlation with the OPC, but the information available of those measurements does not allow for further investigation of the possible causes. Moreover, in the reply to Comment #19 (Referee #1) in this document the differences of particle diameters range measured have been pointed out, leading to a prone overestimation of ACSM with respect to SMPS and therefore a likely reduction of correlation between both instruments.

**44. Table S3: ACSM nitrate being larger than the off-line concentration is quite strange. Usually, off-line instruments have larger cut-offs, even if both are $PM_1$, and sample higher nitrate concentrations than an ACSM. Could that be IE problem? This also relates to the comment on OM RIE, is there only OM problem or other compounds as well, like $NO_3^-$, which would rather point to IE, not RIE?**

One possible explanation for the ACSM overestimation of $NO_3^-$ is the fact that $NO_3^-$ tends to volatilize in filters, as ammonium nitrate is volatile at 25°C, causing a negative artefact, while all $NO_3^-$ mass is measured by the ACSM. This has already been pointed out in many studies (e.g. Ripoll et al., 2015).

Nevertheless, there was an error in the previous Supplementary Table S3, where the slope for $NH_4^+$ ($NH_4^+$ concentration determined by Q-ACSM vs. filter determination) was higher than the $NO_3^-$ one. This has

been corrected now (Table S2) and as expected, the corrected value for the $NH_4^+$ slope is in between the slope of $SO_4^2$ and $NO_3^-$, as $NH_4^+$ is present as a counterion for $SO_4^2$ and $NO_3^-$. The rest of the values were also checked with a more debugged filter samples dataset and a few less significant corrections were applied. The intercomparison figures have now been added to Figure S5 b.

Table S3 has been corrected to:

**Table R2. Regression statistics of ACSM vs. off-line concentrations of $SO_4^{2-}$, $NO_3^-$, $NH_4^+$ and OM-to-OC ratio.**

| (ii) | | A | | | B | | |
|---|---|---|---|---|---|---|---|
| Y | x | $R^2$ | Slope | Intercept | $R^2$ | Slope | Intercept |
| $SO_4^{2-}$ ACSM | $SO_4^{2-}$ off-line | 0.88 | 1.24±0.05 | -0.31±0.10 | 0.93 | 1.05±0.04 | 0.01±0.08 |
| $NO_3^-$ ACSM | $NO_3^-$ off-line. | 0.84 | 1.9±0.1 | -0.12±0.09 | 0.86 | 1.76±0.09 | 0.32±0.09 |
| $NH_4^+$ ACSM | $NH4^+$ off-line | 0.85 | 1.73±0.08 | -0.26±0.07 | 0.71 | 1.68±0.13 | 0.05±0.09 |
| OA ACSM | OC off-line | 0.73 | 2.69±0.17 | -0.9±0.4 | 0.86 | 2.94 ± 0.17 | -1.4± 0.3 |

The corresponding text in Section 3.1 (lines 157-158) was changed too:

*The correlation coefficient for NR-PM$_1$ species of—line analysis of $SO_4^{2-}$, $NO_3^-$ and $NH_4^+$ is always above $R^2>0.71$, and also between OA and organic carbon (OC) ($R^2=0.73$ in A and $R^2=0.86$ in B) (Table S3, Figure S5 b).*

45. **Figure S10: you could get rid of points that are below detection limit for this graph, this would make it clearer.**

As DL values were not available for each one of the seasons of each period, this graph (figure S15 in the present version, previously Figure S10) was eventually filtered by removing the points under the percentile 5 of OA concentrations.

**Anonymous Reviewer #2**

 Review

**Major comments**

1. **The paper is focused on the change of the aerosol chemical composition (included organic aerosol sources) over 4 years. However, the dataset itself covers only the years 1 and 4 of these 4 years. So, in reality, the authors compared the mass fraction and chemical composition only between two years and can therefore hardly conclude on a possible trend since no information on the variation of the aerosol chemical composition and masses between the 2 periods.**

As also the first referee pointed out, the analysis of these two one-year periods cannot be extrapolated as a 4-year trend. Therefore, the approach of this study has been settled now into an inter-year variability, suggesting longer and further studies should seek for a long-term trend. Other variables and pollutants for which the complete time series 2012-2018 is available (such as $NO_x$, $O_3$ and $PM_1$) have been included to help with the interpretation. Moreover, data on previous campaigns on organic sources have been also included to check some key parameters such as LO-OOA-to-MO-OOA ratio in comparison to the ones found in this study. Please see also response to Comment #3 (Reviewer #1) which addresses this same matter.

2.  **The strong limitation for such an approach is to properly quantify the influence of the interannual variability of the meteorological conditions and its impact on the aerosol chemical composition. How did the authors consider this variability? Can their conclusion be biased by the fact that the summer of period A was colder and wetter than a typical summer? Consequently, it is difficult to consider it as an absolute evolution trend over the 4 years.**

The meteorology effects were considered throughout the discussion of the results, also back-trajectories assisted this task. Despite the limitations of the available dataset to be interpreted as a trend, the differences found are assessed with the available information. Whenever possible, full data from 2012-2018 have been provided (see Figure R1, also included in Supplementary as Figure S9), which has given the support for corroboration of descending $PM_1$, BC and OA trends. In addition, from Figure R4, it can be seen how the increase of f43-to-f44 would also suggest an ascend of the SOA oxidation. However, as replied in the previous comment, the focus on the current study has been prudently shifted to the comparison between period A and B, and potential long-term tendencies are only suggested when enough evidence is available.

3.  **Another critical issue is related to the estimation of the chemical composition change between the 2 years itself. The conclusions are based on a direct comparison of the average mass concentration values obtained during periods A and B. For example, the authors did not convince me concerning a possible decrease of the organic mass concentration by 5 % between periods A and B when the respective average concentration at 4.0 (+/- 2.8) and 4.2 (+/- 2.8) μg/m3 (line178) but at the same time this difference is considered as "similar" (line 317). I agree that there is a difference of 5 % between the two values but is this difference significant enough? Could it be rather considered to be inside the instrumental uncertainties or the resulting of the interannual variability? The same comments can be made for all chemical species as well as for the factor analysis.**

As explained in Comment #3 (Referee #1), the approach on the manuscript has been shifted to an interannual analysis rather than a period-wise exploration. Even so, Figure R1 and Table R1 show a decrease for OC along 2008-2017, which indicates a steady reduction of this pollutant. As OA and OC are correlated (as shown in Figure 5b), the same decreasing trend is expected for OA. The same analysis can be extrapolated to EC and BC drops.

4.  **Here a summary table including all average mass concentrations of each species and factors as well as their estimated change including a proper discussion is necessary. Again, in absence of continuous measurement over the 4 years, it is difficult to conclude about a significant change without any statistical analysis for comparing the 2 years.**

The information required has been added to the revised Table S4. The general tone of the manuscript has been modulated to support an inter-annual trend rather than an overall trend. Nevertheless, in Comment #3 (Referee #1), $PM_1$, BC and OC trends have been supported by external measurements.

5.  **Maybe a seasonal approach could also be considered to affine the conclusion of the authors.**

Figure 1 and Figure 4 show the seasonal differences of NR-$PM_1$ species and BC, and OA factors. The authors agree with the reviewer that a seasonal approach is adequate for this dataset, therefore Section 3.2.1 present seasonal variations. In the discussion and the conclusions this issue is addressed considering the highlights of the seasonality analysis.

**6. Moreover, the discussion on the spatial origin of the OA is weakening the argumentation on a possible change in the SOA mass concentration between periods A and B. Line 308, the authors mentioned that the summer of period A was anomalously cold and wet. Could it be that the reported difference of SOA mass concentration between the two periods, been influence by this untypical summer leading to lower SOA formation due to bad weather conditions for period A compare to more typical summer? This is also supported in Figure 8, which shows that LO-OOA and MO-OOA were not coming from the same area during periods A and B.**

It is true that the differences reported between both summers are strongly affected by the meteorological conditions from period A, and this is elaborated in the manuscript. Nevertheless, it might not be the only reason, based on the evaluation of the variations between A and B that follows. The variation in SOA concentration and oxidation state observed between summers A and B is also noticed when assessing monthly averages for the previous and posterior months, which were meteorologically ordinary in both periods (Figure R5). Thus, lower LO-OOA-to-MO-OOA ratios were recorded in B compared to A (1.8 and 1.0, 1.6 and 1.2, 0.9 and 0.8 in May, June and September, and for periods A and B, respectively) (Figure R5). This information was included in the manuscript as follows:

*The sensitivity to meteorological effects on the direct comparison between both summers can be avoided by comparing the LO-OOA-to-MO-OOA ratio between ordinary June and September months (avoiding anomalous and data-lacking July-August) between periods A and B. The degree of oxidation of SOA was also higher in period B (1.6 and 1.2 in June and 0.9 and 0.8 in September).*

[Figure]

**Figure R5. Monthly concentrations of OA factors in bars (left axis) and LO-OOA-to-MO-OOA fraction in dots (right axis).**

A local focus of LO-OOA and MO-OOA can be perceived in both periods, despite the intensity of MO-OOA in A is much lower. This observation supports the fact that the ageing of OOA is more pronounced in period B, as the locally-produced LO-OOA is oxidizing more intensively. Regarding the apparition of, for instance, NNE OOA foci in B not present in A, it could be argued that a new emission focus could have appeared from 2015 and generated these high concentrations towards that direction. However, the observed consistency of LO-OOA and MO-OOA heatmaps stresses the interactivity between these two sources.

**7. Off-line filter analysis: I would expect more details regarding the off-line measurements. What was the sampling time (24h, midnight to midnight)? On the manuscript, only the ICP-AES is mentioned, while the SI referred to ICE-AES and ICP-MS. Could you please clarify? Please also provide the type of the instrument, the manufacturer, and under which condition the analysis was made? A similar comment the can be made for the Ion chromatography analysis and the selective electrode.**

The supplementary Section 2 (in the previous version of the manuscript Section 1) was improved to include all the missing information as follows:

*PM₁ samples were collected on 150 mm-diameter quartz fiber filters using sequential automatic high-volume samplers Digitel (DHA 80, 30 $m^3 \cdot h^{-1}$). The sampling time was 24 hours midnight to midnight and concentrations were assigned to the start date. Sampling frequency was 1 every 3 days. PM₁ levels were determined gravimetrically by conditioning and weighting the filters before and after sampling. Due to problems with the gravimetric determination, PM₁ mass concentration was reconstructed (PM₁ reconstructed) by the addition of all components and an estimation of 25% to account for water.*

*A complete chemical analysis of the collected PM₁ samples was carried out. A quarter of the filter was used for an acidic digestion ($HNO_3{:}HF{:}HclO_4$) following the methodology devised by Querol et al., 2001. The resulting solution was analysed by means of Inductively Coupled Plasma Atomic Emission Spectroscopy (ICP-AES, IRIS Advantage TJA solutions, THERMO) and Inductively Coupled Plasma Mass Spectrometry (ICP-MS, X Series II, THERMO) for the major and trace elements concentration determination, respectively. Few mg of the reference material NIST 1633b were added to a fraction of laboratory blank filters to check the accuracy of the analysis of the acidic digestions.*

*A quarter of the filter was water extracted and the concentrations of $NO_3^-$ , $SO_4^{2-}$, and $Cl^-$ were determined by Ion Chromatography (HPLC) using a WATERS IcpakTN anion column with a WATERS 432 conductivity detector. The concentration of $NH_4^+$ was determined with a Selective Electrode (MODEL 710 A+, THERMO Orion). $SO_4^{2-}$ concentrations in the present study were those calculated from S concentrations determined by ICP-AES, in agreement with all previous works by the research group.*

*A rectangular portion (1.5 $cm^{-2}$) of the remaining filter was used for the analysis of organic carbon (OC) and elemental carbon (EC) by thermal-optical methods using a SUNSET OCEC analyser following the EUSAAR 2 protocol (Cavalli et al., 2010).*

*One blank filter was kept for each set of ten filters. Blank concentrations were subtracted from the total concentration measured for each sample, thus giving ambient concentrations.*

8. **Last but not least, it is known that the value of the OC and EC strongly depend on the reference method used (Chiappini et al., 2014, Cavalli et al., 2010), it is important to provide also the reference of the instrument used for the analysis as well as the applied method (NIOSH-like, EUSAAR-2, IMPROVE_A).**

The EUSAAR-2 protocol was applied (Cavalli et al., 2010) as also specified in Comment #5 (Referee #2). This specification was included in the manuscript:

*Organic Carbon (OC) concentrations were determined in off-line PM₁ samples by thermal-optical methods following the EUSAAR 2 protocol (Cavalli et al., 2010).*

9. **The description and the discussion of the source apportionment analysis need to be improved and to be combined in a single place in the manuscript.**

The description and discussion have been improved in the revised manuscript. Please see response to Comment #6 (Referee #1) for details.

10. **Moreover, the authors included a lot of figures and tables on the supplementary information that are not described at all. In a more general comment, all figures and tables present in a manuscript have to be properly discussed by the authors. This is quite a pity and the description of the factor analysis will certainly gain clarity if the authors described the results of the intermediate factor solution and why they were satisfied by their final factor solution as well as factor identification.**

All figures and tables have been described in the last version of the manuscript. The procedure and criteria followed to select the final solution have been now discussed in the revised version of the manuscript. Please see response to Comment #6 (Referee #1) for details.

**11. For example, the authors considered the presence of a biomass burning factor based on the monthly average of the ratio f60 and f73, what about the ratio of f44 vs f60? Could this average mask some biomass burning events? Did the authors also investigate the presence of BBOA for the other seasons in an early stage of their source apportionment analysis?**

The f44 vs. f60 and f43 vs. f60 plots did not reveal any potential BBOA source in the previous/following months of the selected ones.

Firstly, a seasonal PMF based on meteorological seasons (DJF, MAM, JJA, SON) was performed, anchoring BBOA with two different reference profiles on all of them, but this analysis was discarded as the profile resulted noisy or with very high residuals on f60, f73. Nevertheless, in MAM and SON there was a progressive decay of the f60, f73 (and others BB-related ions) scaled residuals in March and November, as well as reasonable f60, f73 concentrations, also in the f43 vs. f60 plots. For this reason, the decision of extending the winter to November-March was taken and the results showed to be satisfactory in terms of errors and fitting to the time series dynamics.

The text was changed as the 3rd paragraph in the cited text in Comment #6 (Referee #1).

**12. I also have a small concern regarding the averaged values of the BBOA contribution (6% for period A and 4% for period B, figure 2). When comparing the different average values (figures 2, 5, 7, and 9), I got the impression that the BBOA contribution is slightly overestimated compare to what can be expected from figure 5 (it could also be an averaging effect). This is also the case when regarding the BBOA contribution between periods A and B in figure 7, the overall BBOA mass faction seems to be always higher than 10% (especially for period B). Did the authors consider the BBOA mass concentration as 0 μg/m3 or as no value for seasons when BBOA was not identified? Also, a discussion will help to understand the choice of the single OOA factor for period A during the Nov-March period.**

Period-averaged BBOA concentrations were calculated with zeros in the months with no BBOA so that the total yearly concentration was representative in front of the other factors. Hence, while annual average BBOA contributions were 6% and 4% for periods A and B, respectively, the BBOA contributions in the November-March subperiods were 14% and 11%, respectively, for periods A and B (Figure 4 in the revised version, Figure 5 in the previous version). In Figure 7, BBOA relative concentrations were presented taking substituting zeros by NaN (Not a Number), representing the November-March OA contribution so that its change regarding OA would be legible enough.

The reasons why only one OOA factor was chosen for the November-March 2014-2015 period is that in the f44 vs. f43 plots the two OOA factors retrieved were not placed diagonally, suggesting a sufficient differentiation of both profiles, but together in an intermediate space. When 5 factors were forced to appear, the OOA factors just represented a split of an oxidized OOA factor, with similar f43-to-f44 ratios.

**13. How the two mass spectra of this MO-OOA compared to the 3 other MO-OOA of period A?**

In Figure S10 (Figure S3 in the previous version), the resemblance between OOA Nov-Mar 2014-2015 can be observed in comparison to the rest of the period A MO-OOA profiles. For further clarity and rigour, scatterplots of the profiles of MO-OOA compared to the November-March 2014-2015 MO-OOA profile are shown (Figure R6). In all cases, the correlation coefficient is above 0.96 and the f44 concentration near the 1:1 line, thereby the MO-OOA label for this Period A Nov-March OOA factor is sufficiently robust.

[Figure]

**Figure R6.** Comparison of November-March OOA profile with the MO-OOA profiles from the rest of the seasons in period A.

**14. The discussion on the special origin of the aerosol is very interesting but it would be extremely helpful to include a map showing the surrounding of the station when describing for example the presence of roads, or residential areas.**

This map has been included in the Supplementary as Figure S1.

**15. Similarly, the different air mass trajectories analysis is poorly described in the current version of the manuscript. Here again, it would be helpful for the reader to know which method was used to classified the trajectory, to characterize each of them by their frequency of occurrence, their meteorological characterization (main period of occurrence, average temperature, wind speed. . .), and to include a figure with the different class of trajectories.**

The main characteristics of the air masses and the methodology employed to classify them are described in the cited paper (Pey et al., 2010) and the references therein. The same method was applied in the present study. For the sake of brevity, the description was kept as short as possible. Nevertheless, further discussion of the inter-period differences has been added in revised section 2.5.

*Classification of atmospheric episodes was performed with the HYSPLIT model (Stein et al., 2015). Air mass back-trajectories for 120 h at three heights (750, 1500 and 2500 m a.s.l) were computed, with vertical flux modelling, for each day of measurements and interpreted to be classified regarding its predominant transport provenance into Atlantic North (AN), Winter Anticyclonic (WA) (from October to March), Europe (EU), Mediterranean (MED), North African (NAF) and Summer Regional (SREG, from April to September), characteristics of which are discussed in previous works (Pey et al., 2010; Ripoll et al., 2014, 2015). In Figure S3 (a) the relative contribution of each scenario per month is shown. The main difference between period A and B is on summer months, in period B, the main episode is SREG (always >60% of the days) whilst in A its proportion is reduced and more AN and AW episodes take place.*

**16. Last but not least, it is damageable that the inorganics species and BC are not included in the trajectory analysis.**

The trajectory analysis has been performed also for NR-PM$_1$ species and BC and it is included in the Figure S18 b (Figure S13 in the previous version of the manuscript). As not many differences between trajectory scenarios have been depicted, no discussion has been performed in this direction. The following sentence was added to the manuscript to provide reference to Figure S18 b.

*The trajectory analysis was also performed on NR-PM$_1$ species and BC, but no significant features were found regarding episodes (Fig. S18 b).*

**17. It would certainly provide added values to support the discussion. - The statements of the authors must be systematically supported by numbers. Only mentioning "highest", "higher", "lower", "high proportion", "lowest" is hard to understand if there are no clear references.**

This has been revised and specific values have been added to the manuscript for a better quantitative assessment of variations.

**Minor comments:**

**18. Abstract: line 13: is the different between 10.1 and 9,6 µg·m$^{-3}$ statistically significant?**

This significance of these values has been expounded in Comment #3 (Referee #1) and supported by a long-term decreasing trend for this pollutant.

**19. Line 15: please support the term "significant" decrease and "higher" by numbers.**

This sentence was completed as follows:

*Regarding inorganic compounds, SO$_4^{2-}$, black carbon and NH$_4^+$ showed a significant decrease from period A to B (-21%, -18% and -9% respectively), whilst NO$_3^-$ concentration was found higher in B (+8%).*

**20. Line 21: how the authors defined SOA based on their PMF results?**

This was corrected in the manuscript as follows:

*Source apportionment revealed OA was 46% and 70% of secondary origin (SOA) in periods A and B, respectively. Two oxygenated secondary sources (OOA) were differentiated by their oxidation status (i.e. aging): less-oxidized (LO-OOA) and more-oxidized (MO-OOA).*

**21. Line 37: please check the uniformity of the referenced labelling.**

The references are correctly labelled in the revised version of the manuscript.

**22. Line 65 and rest of the manuscript: please choose between Q-ACSM or ACSM.**

The change to Q-ACSM has been applied throughout the whole manuscript.

**23. Line 70: 70 eV.**

This change was applied to the manuscript.

**24. Line 75: Could the authors please precise how often the ACSM was calibrated during each period? Are the IE and RIE a single calibration or an average value?**

The Q-ACSM was calibrated every 4-6 months with the jump scan technique on period A and the full scan in period B. The calibration which offered a best agreement with external $PM_1$ considering both slope and $R^2$ and a proper neutralization of the aerosol was selected.

**25. Line 86: Could the authors precise the range of the OPC measurements and how is it comparing to the SMPS?**

The OPC range is 0.3-10 µm, and the data considered here are 0.3 - 1 µm, although its correlation coefficients were ascertained as untrustworthy and removed from the manuscript and supplementary (see Comment #43 (Referee #1). The SMPS range used is 17.5-478.3 nm. The $PM_1$ mass concentration determined by the OPC was compared with the mass concentrations derived from the SMPS ($R^2$= 0.47 and 0.49 for A and B, respectively). The comparison is not included in the manuscript for the sake of brevity.

**26. Line 93: is the SIR S-5012 $NO_x$ monitor equipped or not with a blue-light converter?**

The instrument used to measure $NO_x$ was Thermo Scientific, Model 43i and it was not equipped with a blue-light converter. The information has been updated in the revised manuscript (the model written in the initially submitted manuscript (SIR S-5012) was incorrect.

**27. Line 93-96: Were all the additional measurements performed at the same place or not?**

All the measurements are co-located to the Q-ACSM in PR except for the meteorology variables, which were measured in the University of Physics, which is 400 m away in SE direction (see map in Fig. S1).

**28. Line 100: Could the authors refer her to section 3.1. At this stage, it is not clear that the comparison will be discussed later on.**

This reference was included in the manuscript.

**29. Line 116 and follow: only ME-2 allows to constrain a factor and the use of the a-value, PMF cannot do that.**

This change was implemented in the manuscript as it has been cited in Comment #6 (Referee #1).

**30. Line 125: Please also include here reference to Canonaco, F., Crippa, M., Slowik, J. G., Prévôt, A. S. H., and Baltensperger, U.: SoFi, an IGORbased interface for the efficient use of the generalized multilinear engine (ME-2) for the source apportionment: ME-2 application to aerosol mass spectrometer data, Atmos. Meas. Tech., 6, 3649-3661, doi:10.5194/amt-6-3649-2013, 2013.**

This reference was included in the text when mentioning SoFi as presented in Comment #6 (Referee #1).

**31. Lines 121, 128: The authors should define their acronym the first time they use them in the manuscript (here BBOA, HOA, COA), not a few lines later (here line 138-140).**

This was changed so that the definition of factors acronyms appears now in the introduction section, which has been exposed in Comment #4 (Referee #1).

**32. Line 153: The results of the comparison between ACSM and co-located measurements will gain an understanding by also including the plots on the SI and not only a table.**

These plots were included as Fig S6 in the supplementary section.

**33. Line 153 and Table S3: please check the homogeneity of the fitting parameters. The authors used either one or three digits for the same analysis. Only, 2 digits are necessary here.**

The number of digits shown for each parameter were based on the associated uncertainty. If the first significant digit of the uncertainty cypher is the third decimal, the parameter value should be 3 decimals long too, hence showing significant digits.

**34. Line 157: The authors should not only look at the $R^2$ to consider that the agreement is "very good" but also on the slopes and intercepts. For example, nitrate and sulfate correlation have both high $R^2$ (0.84 to 0.93, respectively) but their slopes (1.9 for the nitrate in period A and 1.32 for sulfate in period B) show a large overestimation by the ACSM. This must be discussed too before concluding on a "good agreement".**

The discussion has been improved by digging further on the slopes over 1 specially for $NO_3^-$ and $NH_4^+$ as presented in Comment #20 (Referee #1). The overestimation of the Q-ACSM cannot be discarded, although slopes with other $PM_1$ measurements are not unequivocally supporting this hypothesis.

**35. Line 160: I do not fully agree with the conclusion of the authors. The comparison between the organic mass measured by the ACSM and the OC from the filters can also be impacted by the reference method used for the OC estimation. Did the authors try to convert their organic mass into OC directly compared to the offline OC?**

The OC concentration determined in the filter samples was carried out by thermal-optical method following EUSAAR 2 protocol as described in the Section 2 (previously Section 1) of Supplementary. The determination of the slope from the linear regression between OA from the Q-ACSM and OC from filter samples has been interpreted as the OM-to-OC ratio, resulting in 2.69 and 2.94 in periods A and B respectively. These values from the current study do not match the value of 1.6 determined in Mohr et al. (2012) using an AMS during the DAURE campaign, in March 2009. Obviously, the OM-to-OC ratio could have changed over the years since March 2009, hence expecting an OM-to-OC ratio higher than 1.6, which would agree with the higher proportion of SOA found in these campaigns (2014-2015 and 2017-2018) compared to DAURE. Nevertheless, values over approx. 2.3 are not expected for OOAs according to both chamber and ambient measurements, as the most oxidized existing organic compounds, such as glycolic acid or oxalic acid have OM-to-OC ratios of approx. 3. Therefore, 2.69 or 2.94 values found in this study (when comparing OA from Q-ACSM and OC from filter samples) exceed the expected ratios. This consideration has now been included in the revised manuscript. A possible source of disagreement in the OM-to-OC ratios could be related to filter artefacts, which cannot be discarded. Moreover, OM-to-OC was estimated from f44 as stated in (Canagaratna et al., 2015), bearing in mind that this should be interpreted with a note of caution, as deriving OM-to-OC from f44 from Q-ACSM could lead to significant instrument-dependent discrepancies due to f44 variability amongst them as stated in Fröhlich et al. (2015). Nevertheless, the relative variation of this value can be considered valid, when it is determined with the same instrument in the same location and within a reasonable period of time, as it is the case of the present study. The monthly mean values for this OM-to-OC ratio are presented in Figure R7. In spite of the higher values in winter of Period A, it can be seen how the values for warm months are always higher for the Period B, implying a potentially

increasing degree of oxidation from period A to period B. This could entail that the OM-to-OC ratio could have effectively changed since 2009, and definitely from 2014-2015 to 2017-2018. However, again, when interpreting OM-to-OC ratios from Q-ACSM compared to AMS, this comparison should be done with caution, given the limitations stated above for Q-ACSM (Fröhlich et al., 2015).

[Figure]

**Figure R7. Monthly averages of OM-to-OC ratio for periods A and B estimated as in Canagaratna et al. (2015).**

The manuscript was modified as follows:

*OA-to-OC ratio is estimated from the slope in the scatterplot between these two variables, resulting in values of 2.69 and 2.94 in periods A and B, respectively, a 68% and 84% higher than the 1.6 value calculated with an AMS in Barcelona in the DAURE campaign (Minguillón et al., 2011; Mohr et al., 2015). Although the OM-to-OC ratio might have increased over the years since March 2009, which would be in accordance to an increasing SOA proportion found in these campaigns (2014-2015 and 2017-2018) respect to DAURE. The values found in the present study are too high, as the expected OOA values according both to chamber and ambient experiments should be around 2.3 (Canagaratna et al., 2015). This too high OA-to-OC ratios would point to filter artefacts caused by evaporation of semi-volatile OC compounds as well as similarly to the previous findings at Montsec (Ripoll et al., 2015) and Montseny (Minguillón et al., 2015), the aforementioned unspecific-source OA RIE overestimation in Q-ACSM. Even so, a steady increase of OOA oxidization over the years could be inferred due to the growth of this ratio from 2014-2015 to 2017-2018.*

**36. Line 165: why not including the time series of the ACSM measurements directly in the main manuscript.**

The full time series of the ACSM measurements result in a too-crowded plot and the information retrieved from it is limited; this is the reason to have it in the supplementary material. Moreover, the time series plots of the OA sources have also been moved to the Supplementary section as requested by Referee #1.

**37. Line 262: Is the number of "high", not a bit too "high" in this sentence?**

This sentence was rephrased as follows:

*However, concentrations of SOA are simultaneously elevated with high temperature and CO levels, …*

**38. Line 291: reference to Fig. 8 is missing?**

This reference was now included in the manuscript.

**39. Line 297: could you please add a reference here?**

The reference was added in this position: (Toll and Baldasano, 2000).

**40. line 319: Please refer to the corresponding figure.**

The figure reference has been added.

**41. Figure 2: Should the total mass of organic not be 4.2 µg/m3 for period B?**

This graph shows the apportioned OA and the contribution of their factors. The OA concentration here is the sum of all OA factors, therefore due to the error of PMF, it is mismatching with the 4.2 µg/m3 which represents the measured OA.

**42. Figure 8: The legend of the color code is missing.**

The variable of the legend of colour of Figure 8a (now Figure 7) was added. The caption has also been improved:

*Figure 7. Wind dependence of OA factors. (a) Windrose plots showing frequency of counts (in percentage) regarding wind direction and wind speed (m·s$^{-1}$). (b) Polar plots color-coded by median mass concentrations (µg·m$^{-3}$) of OA factors listed below. In both cases, plots are arranged to the left for Period A and to the right for Period B.*

**43. Figures 3,4,6, S1, S2, S3, S4, S5, S9, S12: legend of the x-axis is missing.**

These figures were changed and the x-axis included.

**44. Figure S8: The legend of the y-axis is missing.**

This figure axis was included.

**45. Diurnal figures: It is hard to identify the different seasons as well as the hours on the present figures. I would strongly suggest separating each diurnal profiles.**

This figure has been improved and now the diels are separated by seasons (Figure S13).

**46. Supplementary section 1: please provide more details regarding the sampling regime, the analytical instrumentation, and methods used.**

The details have been provided in the Supplementary section 2 (Section 1 in the previous version of the supplementary). Please see details in reply to Comment #7 (Referee #2).

**47. Tables on the supplementary information: Please reorganize the tables to have a complete table on one page.**

This change was applied.

---

## Author Response (AR2)

**Reply to reviewers II (acp-2020-1244)**

The authors would like to thank the reviewers for their minor comments and suggestions, which helped improving the quality of this work. Once more, a new version of the manuscript has been prepared following the suggestions from the reviewers and replies for each comment are addressed in a point-by-point manner.

**Anonymous Reviewer #1**

I commend authors for such thorough and detailed revision that has addressed all my concerns. I recommend the manuscript for publication after few technical corrections listed below, are maid:

1. **Title: I do not think that 'interannual increase' is a correct term, 'interannual variation', maybe? Or, since you did provide a better proof for the trend, you could refer to an increase in secondary aerosol fraction due to potential growth in atmospheric oxidation potential, just a suggestion.**

The title was changed as follows:

*Increase of secondary aerosol fraction in an urban background.*

2. **Reply to comment No 4: 'In Zurich, for instance, f44 during summer afternoons, when photochemical processes are most vigorous as indicated by high oxidant – OX (O3 + NO2) was found similar or lower than f44 on days with low -OC, while f43 (less-oxidized fragment) tended to increase (Canonaco et al., 2015). The SOA is often divided into two factors: less oxidized oxygenated OA (LO-OOA) and more oxidized ozygenated OA (MO-OOA)'. Is this really correct - f44 lower for days with higher oxidation? Or this is a mistake, and the opposite was meant?**

This sentence is in accordance with the findings from (Canonaco et al., 2015), although it was rephrased for better understanding as follows:

*In Zurich, however, f44 during summer afternoons, when photochemical processes are most vigorous as indicated by high oxidant – OX ($O_3$ + $NO_2$), was found similar or lower than f44 on days with low – OX, while f43 (less oxidized fragment) tended to increase (Canonaco et al., 2015). The SOA, also referred as Oxygenated OA (OOA), is often divided into two factors: less-oxidized oxygenated OA (LO-OOA) and more-oxidized oxygenated OA (MO-OOA).*

3. **Reply to comment No 7: 'There are evident differences in meteorological variables from one period to the other (Fig. S7). In summer A, temperature (24.0° C) and solar radiation (259 W) are lower and average relative humidity (71%) and wind speed (2.0 m/s) are higher than in period B (27° C, 280W, 70.0%, 1.7m/s), indicating a probably rainier or cloudier summer in period A.' Check units for B, I assume 1.7 is m/s? Also,**

**are such differences in RH (70% and 71%) as well as WS (2 vs. 1.7 m/s) significant? Otherwise, just state that they were similar.**

The units of the meteorological variables for period B were corrected as follows:

*(27.0º C, 280W, 70%, 1.7m/s)*

Neither relative humidity (RH) nor wind speed present significant variations. In the case of the relative humidity, a month-by-month Mann-Kendall test revealed a decreasing trend, which would be in line with the slight decrease that shows the period-average figures. The same statistic for wind speed does not support the decrease from period A to B, so the text was modified as follows:

*There are not significant differences in meteorological variables from one period to the other (Fig. S7). In summer A, temperature (24.4ºC) and solar radiation (259 W) are lower, and relative humidity and wind speed (2.0 m/s) is similar compared to period B (27.0º C, 280W, 70%, 1.7m/s, respectively). These results are in line with a colder and cloudier summer in A as is depicted in* Meteocat (2020a, 2020c, 2020b).

4. **Reply to comment No 7: 'The composition-dependent collection efficiency (CE) correction (Middlebrook et al., 2012) correction was applied (minimum, maximum) for Period A and B respectively; (0.45±0.68), (0.50, 0.99), exceeding CE=0.6 a 0.13% and a 1.5% of data).' Is period B data missing here? I see the average, SD and range for the A period only?**

The authors were referring to minimum and maximum values, therefore, not averages nor SD were provided. The actual text was written as follows:

*The composition-dependent collection efficiency correction (CE) (Middlebrook et al., 2012) was applied for the two periods with minimum and maximum values of 0.45 and 0.68 for period A and 0.50 and 0.99 for period B, exceeding CE=0.6 a 0.13% and a 1.5% of data, respectively.*

**Anonymous Reviewer #2**

The authors did remarkable work in revising their manuscript and taking care of the comments which were properly addressed.

1. **However, I still have some concerns regarding the conclusion which does not contain the main message of the work and appears poorly structured.**

The conclusion was restructured as follows:

*Characterization of non-refractory fine aerosol (NR-PM$_1$) in the urban background of Barcelona including organic aerosol (OA) source apportionment was performed regarding two nearly one-year periods between 2014 and 2018. Period-averaged PM$_1$ concentrations were 10.1 and 9.6 µg·m$^{-3}$ respectively for the so-called periods A (May 2014 - May 2015) and B (September 2017 - October 2018). Slopes between total mass concentration of Q-ACSM species and BC and PM$_1$ retrieved by a Scanning Mobility Particle Sizer were near one, but Q-ACSM overestimation caused by the use of the default relative ionization efficiency for OA, probably lower than the actual one, cannot be discarded.*

*Average contributions of the inorganic NR-PM$_1$ were 19% and 18% for SO$_4^{2-}$, 16% and 13% for black carbon (BC), 10% and 13% for NO$_3^-$, 11% and 11% for NH$_4^+$ and <1% for Cl$^-$ for periods A and B, respectively. Hence,*

*$SO_4^{2-}$ and BC concentrations decreased from A to B, while $NO_3^-$ ascended. Seasonal NR-PM$_1$ cycles consist of the maximization of OA and $SO_4^{2-}$ in the warmer subperiods and $NO_3^-$ (high volatility in hot conditions) and BC in the coldest. $NO_x$ were also reduced on average from A to B, as well as $O_3$ and $SO_2$. Nevertheless, $O_3$ became more reactive on period B.*

*The OA was the major component in both periods under study, accounting for a 42% of total PM$_1$ in both periods. Five organic sources were identified by PMF: Cooking-like OA (COA), Hydrocarbon-like OA (HOA), Biomass Burning OA (BBOA), Less-Oxidized Oxygenated OA (LO-OOA) and More-Oxidized Oxygenated OA (MO-OOA). BBOA was only present in the subperiod November-March and only one OOA (Oxidized OA) factor was apportioned in this cold subperiod in 2014-2015. Secondary OA (SOA, comprised by the sum of OOAs) proportion and absolute concentrations increased from the first period to the second, while primary OA (POA) concentrations were reduced. In turn, LO-OOA and MO-OOA changed its relevance in OOA contributions, being the most oxidized the promoted one from 2014-2015 towards 2017-2018. The oxidation could be partially attributed to the action of solar radiation and temperature, in warm months both higher in period B, as the ratio LO-OOA-to-MO-OOA is also lower in summer. Nevertheless, the decreasing trend of $NO_x$ and the increase of $O_3$ reactivity would position the increasing potential oxidation as a feasible cause as many other urban studies have reported. Seasonal variation of OA contributions was also affected by air masses origin. Northern flows and stagnation episodes induced primary pollution events, although high-SOA events were driven by long-range episodes, comprising Mediterranean and European advections (mainly MO-OOA), and regional breeze-driven recirculation (mainly LO-OOA).*

*Studies of NR-PM1 chemical composition and OA sources have been performed extensively in urban background sites, although to the authors' knowledge, this is one of the first in which the approach benefits of interannual comparison and airmass origin determination in the western Mediterranean basin. The results obtained highlight the role of SOA as the main source of OA and its permanency even if POA is reduced in this site. Moreover, the most aged-SOA airmasses are related with long-range European and Mediterranean circulations which drag pollutants accumulated during several days. However, some gaps remain unsolved about the oxidation of the urban-background atmosphere of Barcelona such as possible pathways of production and transformation of SOA and their interconnection with $O_3$ processes could be the objective of further investigations.*

Minor comments:

2. **The authors must reconsider the notation of their fitting parameters. Do they need to include 4 digits like lines 198-199 and Tab S1, while only 2 digits are used for the rest of the manuscript?**

In the new version of the manuscript, the decimal digits are set to two in all of the squared correlation coefficient mentions, including Tables and Figures.

3. **Line 433: replace "hydro-carbon-like" with "hydrocarbon-like".**

This change was implemented on the manuscript.

4. **Fig. S3: The three different colors of grey are not easy to distinguish.**

These figures were improved for the sake of readability as follows:

[Figure]

[Figure]

**5. Figure S5: It would be better if all axes intercepted at zero.**

Figure S5 was modified according on the reviewer's suggestion.